# Disentangling *Crocus* Series *Verni* and Its Polyploids

**DOI:** 10.3390/biology12020303

**Published:** 2023-02-14

**Authors:** Irena Raca, Frank R. Blattner, Nomar Espinosa Waminal, Helmut Kerndorff, Vladimir Ranđelović, Dörte Harpke

**Affiliations:** 1Department of Biology and Ecology, University of Niš, 18000 Niš, Serbia; 2Leibniz Institute of Plant Genetics and Crop Plant Research (IPK), D-06466 Seeland-Gatersleben, Germany; 3German Centre of Integrative Biodiversity Research (iDiv) Halle-Jena-Leipzig, D-04103 Leipzig, Germany; 4São Romão, Cx 548 A, 8150-058 São Brás de Alportel, Portugal

**Keywords:** chromosome numbers, *Crocus heuffelianus* group, *Crocus* series *Verni*, dysploidy, genome size, genotyping-by-sequencing (GBS), morphometry, phylogeny, polyploidy

## Abstract

**Simple Summary:**

In plants, the occurrence of polyploid lineages, which are plants with multiple instead of two sets of chromosomes, is quite common. Polyploids can originate as autopolyploids within a species or by combining the genomes of different species resulting in allopolyploids. Within the group of spring crocuses, a polyploid complex exists where it is unclear how it evolved and which species eventually contributed to polyploid formation. Among *Crocus* species, evolutionary analyses are further complicated by widely varying chromosome numbers that do not clearly correlate with di- or polyploidy. To reconstruct the evolution of these polyploids, we combine chromosome counts, genome size estimations, phylogenetic analyses based on maternally and bi-parentally inherited genomes, co-ancestry analysis, and morphometric data for all species potentially involved in polyploid formation. Through this approach, we show that polyploids in the *Crocus heuffelianus* group are allopolyploids that originated multiple times involving different parental genotypes and reciprocal crosses. Chromosome numbers partly changed after polyploidization. Numbers found in polyploids are therefore no longer in all cases additive values of their parents’ chromosomes. We conclude that in crocuses, only an approach combining evidence from different analysis methods can uncover the evolutionary history of species if polyploidization is involved.

**Abstract:**

Spring crocuses, the eleven species within *Crocus* series *Verni* (Iridaceae), consist of di- and tetraploid cytotypes. Among them is a group of polyploids from southeastern Europe with yet-unclear taxonomic affiliation. Crocuses are generally characterized by complex dysploid chromosome number changes, preventing a clear correlation between these numbers and ploidy levels. To reconstruct the evolutionary history of series *Verni* and particularly its polyploid lineages associated with *C. heuffelianus*, we used an approach combining phylogenetic analyses of two chloroplast regions, 14 nuclear single-copy genes plus rDNA spacers, genome-wide genotyping-by-sequencing (GBS) data, and morphometry with ploidy estimations through genome size measurements, analysis of genomic heterozygosity frequencies and co-ancestry, and chromosome number counts. Chromosome numbers varied widely in diploids with 2*n* = 8, 10, 12, 14, 16, and 28 and tetraploid species or cytotypes with 2*n* = 16, 18, 20, and 22 chromosomes. *Crocus longiflorus*, the diploid with the highest chromosome number, possesses the smallest genome (2C = 3.21 pg), while the largest diploid genomes are in a range of 2C = 7–8 pg. Tetraploid genomes have 2C values between 10.88 pg and 12.84 pg. Heterozygosity distribution correlates strongly with genome size classes and allows discernment of di- and tetraploid cytotypes. Our phylogenetic analyses showed that polyploids in the *C. heuffelianus* group are allotetraploids derived from multiple and partly reciprocal crosses involving different genotypes of diploid *C. heuffelianus* (2*n* = 10) and *C. vernus* (2*n* = 8). Dysploid karyotype changes after polyploidization resulted in the tetraploid cytotypes with 20 and 22 chromosomes. The multi-data approach we used here for series *Verni*, combining evidence from nuclear and chloroplast phylogenies, genome sizes, chromosome numbers, and genomic heterozygosity for ploidy estimations, provides a way to disentangle the evolution of plant taxa with complex karyotype changes that can be used for the analysis of other groups within *Crocus* and beyond. Comparing these results with morphometric analysis results in characters that can discern the different taxa currently subsumed under *C. heuffelianus.*

## 1. Introduction

Polyploidization or whole-genome duplication (WGD) is a common process in plants, resulting in individuals with different ploidy levels. Two major mechanisms are discerned: if WGD happens within a species, the resulting polyploid is termed autopolyploid, whereas hybridization between two different species followed by WGD is termed allopolyploidy [1]. Autopolyploids might suffer at least initially from reduced fertility due to distorted chromosome distribution during meiosis, while allopolyploids usually undergo normal meiosis. However, one has to understand auto- and allopolyploids as the endpoints of a continuum. Chromosome pairing and distribution among daughter cells depends on the overall similarity of the chromosomes. Thus, even within a species, chromosomes can be relatively different or rather similar in an allopolyploid if closely related species are involved. Polyploidization is often a driver of evolution, as doubling of genes releases one of the homeologous copies from purifying selection, allowing it to obtain new functions [2]. This event should be advantageous, particularly in stressful habitats or during changing environmental conditions [3]. Through time, polyploid genomes will accumulate differences not only at the level of allelic differences but, through karyotype changes, also regarding the overall structure of the genomes [1]. This results in diploidization, i.e., former homeologous chromosomes are no longer recognizable as such. Ancient polyploidization events are therefore not easy to detect and most often need in-depth genome analysis to be revealed [2,4]. For more recent WGD events, chromosome numbers and genome sizes can be indicative. While there are taxa where chromosome numbers correlate clearly with ploidy levels and genome size [5,6], dysploid chromosome number changes can blur such correlations through breaking and/or fusion of chromosomes [7]. In addition, downsizing or enlarging of genomes [4,8,9,10] through the loss of DNA or activation of transposable elements can hinder ploidy level recognition. However, these latter processes are normally acting at a slower pace than changes in chromosome numbers [11,12].

*Crocus* series *Verni* B.Mathew is a group of mostly spring-flowering crocuses from Central and South Europe, some of them being important ornamentals. The series consists of eleven species with unclear phylogenetic relationships, as earlier approaches with molecular markers arrived only at badly resolved species groups and species identification was also partly uncertain [13,14,15] or sampling incomplete [16]. Chromosome numbers range from 2*n* = 8 to 2*n* = 28 [16,17,18,19,20,21,22,23], often with uncertain ploidy levels, as high chromosome numbers do not necessarily correlate with larger genome sizes and, therefore, higher ploidy level [16]. Particular populations from the Balkan Peninsula, thought to taxonomically belong to *C. heuffelianus* Herb., exhibit highly diverse chromosome numbers with 2*n* = 10, 18, 19, 20, 22, and 23 chromosomes [16,18]. Harpke et al. [15], in their account of series *Verni*, concluded from karyotype analysis that certain populations of *C. heuffelianus* (as well as *C. neglectus* Peruzzi and Carta) might have resulted from polyploidization but were not able to confirm this further. We refer to these potential polyploids throughout this article as “*C.* cf. *heuffelianus*”, “*C.* cf. *tommasinianus*”, and “*C.* cf. *vernus*”, as their taxonomic status is unclear and name changes seem still premature to us. *Crocus neglectus,* in contrast, was already recognized as a separate species [15].

Here we intend to understand the evolution of *Crocus* series *Verni* with particular reference to the origin of the polyploid species and cytotypes. To arrive at this goal, we use molecular phylogenetic approaches based on (a) genotyping-by-sequencing (GBS [24]) to obtain highly informative genome-wide single-nucleotide polymorphisms (SNPs) for a robust phylogeny, (b) nuclear rDNA internal transcribed spacers (ITS) plus 14 single-copy gene sequences for tracing bi-parentally inherited genome parts and chloroplast DNA sequences for inferring maternal lineages within the study group, and (c) morpho-anatomical analyses to find traits that can discern the taxa, and combine these data with (d) chromosome counts and (e) genome size estimations of diverse populations in order to infer di- and polyploid taxa and cytotypes and reveal their parental contributors and geographic distribution.

## 2. Materials and Methods

### 2.1. Plant Materials

Our study includes plants from 63 populations: 24 *C. heuffelianus*/*C.* cf. *heuffelianus*, 13 *C. vernus* (L.) Hill/*C.* cf. *vernus*, nine *C. tommasinianus* Herb., three *C. bertiscensis* Raca, Harpke, Shuka, and V.Randjel., three *C. neapolitanus* (Ker Gawl.) Loisel., two *C. neglectus*, two *C. kosaninii* Pulević, one of each of the other series *Verni* species, two populations of outgroup *C. malyi* Vis., and one ornamental cultivar (Table 1, Appendix A). To differentiate the 2*n* = 18 karyotypes of *C.* cf. *heuffelianus*, we labeled them Western Carpathian clade (WCC), Pannonian-Illyric clade (PIC), and Southern Carpathian clade (SCC). The sampling covers the whole distribution area of the series except for the westernmost *C. vernus* populations from the Pyrenees, which could only be included in the chloroplast and nuclear single-copy marker dataset (Table 1). To encumber poaching on the wild populations we here provide only rather general locations for the studied materials instead of populations’ GPS coordinates. Chloroplast and nuclear single-copy markers were investigated for a smaller number of representatives of series *Verni* while the GBS analyses were based on the most exhaustive number of individuals (Table 1).

### 2.2. DNA Extraction and Sanger Sequencing

Total genomic DNA was extracted from silica-gel-dried leaf tissue with the DNeasy Plant Mini Kit (Qiagen) according to the instructions of the manufacturer. After DNA extraction, we checked DNA quality and concentration on 1% agarose gels. For the amplification of the two chloroplast regions, we used the primers matKf, rpS16in1_r, rpS16in1_f, trnQr, ycf1bF, and ycf1bR [25]. PCR amplification protocols for all markers followed Harpke and Kerndorff [25]. Forward and reverse strands of both regions were directly Sanger sequenced on an ABI 3730 XL using the amplification primers, edited where necessary, and assembled into single sequences in Geneious Prime 2022.1.1 [26]. Afterward, the sequences were aligned using Mafft version 1.5.0 [27] within Geneious and manually corrected.

### 2.3. Genotyping-by-Sequencing

To obtain genome-wide SNPs, GBS analyses [24] were conducted for 91 di- and 102 tetraploid individuals, with one of the latter included twice as a replicate. For the library preparation, 200 ng of genomic DNA was used and cut with the two restriction enzymes *Pst*I-HF (NEB) and *Msp*I (NEB). Library preparation, individual barcoding, and single-end sequencing on the Illumina NovaSeq were performed following Wendler et al. [28].

Barcoded reads from the 194 samples were de-multiplexed using the Casava pipeline 1.8 (Illumina). Adapter trimming of GBS sequence reads was performed with Cutadapt [29] within ipyrad v.0.9.58 [30] and reads shorter than 60 bp after adapter removal were discarded. GBS reads were clustered using the ipyrad 0.7.5 [30] pipeline with a clustering threshold of 0.85. We tested diverse ipyrad settings but at the end the default settings of parameter files generated with ipyrad were optimal for the other parameters. We generated one output that included the outgroup *C. malyi*, which was used for phylogenetic analyses, and a second output without *C. malyi*, which was used for principal component analysis (PCA) and population assignment analyses.

### 2.4. Analyses of Population Structure

Principal component analysis (PCA) was conducted in ipyrad. For model-based Bayesian population assignment analysis we used the R package LEA [31]. Population assignment was performed for K = 1–15 with 20 repetitions each and ploidy set to four. Additionally, fastStructure [32] was used with “simple” as prior for K = 1–15 with 20 repetitions each for cross-validation. The optimal K was then determined with the function “chooseK” in fastStructure. The Q-matrices obtained with LEA (for K = 5, K = 8) and fastStructure (for K = 4), which include the ancestral assignment frequencies, were sorted using the R package tidyverse [33] and plotted with ggplot2 [34], discerning different ancestral clusters with color-coding. The ggplot2 package was also used for plotting the PCA results.

### 2.5. Heterozygosity and Fst Determination

The allelic constitution of SNP positions was checked in the vcf file obtained with ipyrad to infer if and to what extent at these positions more than two alleles for a heterozygous SNP were present in the diploid individuals. As overall only less than 3% of SNPs were not biallelic, DnaSP v. 6 [35] was used to infer the heterozygosity as well as the fixation index (F*st*) of the data based on the vcf files generated with ipyrad for di- and polyploid individuals. The DnaSP output was used to calculate the ratio of heterozygous sites to the total number of sites of the samples.

### 2.6. Next-Generation Sequencing of Nuclear Markers

Fourteen nuclear single-copy markers were amplified (see Appendix A) using the Phusion High Fidelity DNA polymerase (ThermoFisher Scientific). To obtain sequences of the nuclear rDNA internal transcribed spacer region (ITS), we used the primers ITS-A and ITS-B [36] following the protocol of Blattner [37]. PCR products of each amplicon of one sample were pooled together regardless of their concentration, purified using a NucleoFast plate (Marcherey-Nagel), and finally diluted in 34 µL triple-distilled water. Sixteen microliters of the purified amplicon pool were digested using the NEBNext™ dsDNA Fragmentase kit (NEB) for an incubation time of 1.5 min and another 16 µL of the amplicon pool was digested for 4.5 min following the manufacturer’s instructions (NEB). Both were pooled, 100 µL triple distilled water was added, and a NucleoFast plate (Marcherey-Nagel) was used to remove too-small fragments and contaminants. Fifty microliters were subjected to size-selection targeting fragment sizes of 400–600 bp using BluePippin (Sage Science), blunt-end repaired, and used for sequencing library preparation according to the Illumina TruSeq DNA library protocol. Adaptors and barcodes were ligated to the samples. The libraries were size-selected with BluePippin. Fragment size distribution and DNA concentration were evaluated on an Agilent BioAnalyzer High Sensitivity DNA Chip and using the Qubit DNA Assay Kit in a Qubit 2.0 Fluorometer (Life Technologies). Finally, the DNA concentration of the libraries was checked with a quantitative PCR run. Cluster generation on Illumina cBot and paired-end sequencing 2 × 250 bp on the Illumina HiSeq 2000/2500 and NovaSeq6000 platform, respectively, followed Illumina’s recommendation and included 1% Illumina PhiX library as internal control. The targeted output per sample was 300,000 reads. Reads were initially iteratively mapped against the forward primer as reference or already existing sequences of the marker in Geneious Prime 2022.1.1 using the Geneious mapping tool. In a second step, the pre-defined reads were assembled into haplotypes, i.e., representing the different alleles present in the sequenced marker region. Finally, three out of 14 nuclear single-copy markers (*orcf*, *rcf*2, *topo*6) were informative and were used for the investigation of haplotype differences of di- and polyploids to determine the parental species involved in allotetraploid formation.

### 2.7. Chromosome Counts

Chromosome counts were either obtained from the literature, obtained from direct observations in this study, or extrapolated for a few individuals based on the genome size data together with published chromosome counts of nearby populations. For direct observations, roots were cut about 2 cm from the tips, pretreated with 2 mM 8-hydroxyquinoline for 5 h at room temperature, and then kept in cold water overnight in a refrigerator. The roots were fixed in Carnoy’s solution (3:1 ethanol:acetic acid) for 24 h and stored in 70% ethanol until use. Slide preparation was carried out according to Waminal et al. [38] and Rodríguez-Domínguez et al. [39]. Slides were fixed in 2% formaldehyde solution (47608, Sigma-Aldrich) for 3 min and dehydrated in an ethanol series (70%, 90%, 99%). Chromosomes were stained with 1 µg/µL DAPI in 2× SSC. Images were captured using a 100× objective of an Olympus BX61 fluorescence microscope (Olympus).

### 2.8. Genome Size Measurements

Genome sizes were measured for 134 individuals. Due to a lack of material, we could not measure the genome sizes of *C. neapolitanus* and *C. siculus*. Genome sizes for *C. bertiscensis* were partially taken from Raca et al. [16].

Genome size was determined using propidium iodide (PI) as a stain in flow cytometry with a Cyflow Space (Sysmex Partec) flow cytometer, following essentially the procedure described in Jakob et al. [6]. We mainly used rye (S*ecale cereale*; 16.01 pg/2C) or pea (*Pisum sativum*; 9.09 pg/2C) as internal size standards and the buffer CyStain PI Absolute P (Sysmex Partec). Genome size measurements aimed at identifying diploids and polyploids. To link the genome sizes with the molecular data, we used silica-gel-dried leaves from the same individual used for DNA extraction whenever possible. Initial tests showed that fresh and silica-gel-dried materials arrived at the same genome size estimations in *Crocus*. However, the quality of data obtained is slightly lower for dried leaves. A detailed overview of the material measured and standards used is given in Appendix A.

### 2.9. Phylogenetic Inferences and Origin of Allopolyploids

Maximum parsimony (MP) analyses were conducted in Paup* 4.0a169 [40] using a two-step heuristic search as described in Blattner [37] with 1000 initial random addition sequences (RAS) restricting the search to 25 trees per replicate. The resulting trees were afterwards used as starting trees in a search with maxtree set to 10,000. To test clade support, bootstrap analyses were run on all datasets with re-sampling 1000 times with the same settings as before, except that we did not use the initial RAS step. Paup* was also used to infer the best-fitting model of sequence evolution for the sequence datasets using the Bayesian information criterion (Table 2). Analyses were run for (a) a dataset consisting of the combined sequences of the two chloroplast marker regions *mat*K-*trn*Q and *ycf*1; for the GBS-derived sequences including (b) the diploid species and (c) the diploid plus all tetraploid individuals of the *C. heuffelianus*/*C. vernus* complex plus *C. neglectus*, the only tetraploid species formally recognized to date; (d) for three nuclear single-copy genes; and (e) for the rDNA ITS region. The nuclear single-copy and ITS datasets were used to identify homeotic alleles in tetraploids and their diploid parents instead of deriving detailed phylogenetic relationships. The MP analysis of GBS data derived from di- and tetraploid individuals was used to infer the closest diploid parent of allopolyploids and to see if allopolyploids are monophyletic or originated multiple times.

For the GBS-derived data, we also calculated SVDquartets in Paup*, evaluating all quartets for the dataset consisting of 92 diploid individuals running 500 bootstrap re-samples. Individuals were partitioned according to their species affiliation, and trees were selected using QFM quartet assembly and the multispecies coalescent (MSC) as tree model.

Bayesian phylogenetic inference (BI) was conducted in MrBayes 3.2.7 [41] for the chloroplast, the nuclear single-copy genes, and the ITS datasets. In BI, two times four chains were run for 5 million generations specifying the respective model of sequence evolution. A tree was sampled every 500 generations. Converging log-likelihoods, potential scale reduction factors for each parameter, and inspection of tabulated model parameters in MrBayes suggested that stationary had been reached in all cases. The first 25% of trees of each run were discarded as burn-in.

In addition to the phylogeny-based inference of parental species of allotetraploids we also used Stacks v2.55 [42] to generate an input file for RADpainter [43]. A locus needed to be present in 80% of the individuals of a population and in 50% of all individuals to be processed. Population structure was inferred using 10,000 burn-in steps in the Monte Carlo Markov Chain (MCMC) analysis with 100,000 further iterations, keeping every 1000th sample. This run was continued, adding an additional 100,000 steps, treating the original run as burn-in. To obtain the best posterior state for the tree, 1000 attempts were used. Results were visualized in R with the functions provided with RADpainter.

The GBS-related Appendix A are available online through the e!DAL PGP data repository (https://doi.org/10.5447/ipk/2023/5, accessed on 10 February 2022).

### 2.10. Morpho-Anatomical Analyses

The morphological analysis was performed on fresh material, including 435 individuals in total (*C. vernus*: five populations; *C. heuffelianus*: two populations; mixed *C. heuffelianus*/*C.* cf. *heuffelianus* 2*n* = 18 SCC: two populations; *C.* cf. *heuffelianus* 2*n* = 18 SCC: three populations; *C.* cf. *heuffelianus* 2*n* = 18 WCC: two populations; *C.* cf. *heuffelianus* 2*n* = 18 PIC: two populations; *C.* cf. *heuffelianus* 2*n* = 20: four populations; *C.* cf. *heuffelianus* 2*n* = 22: two populations). Leaf cross-sections were made using a manual microtome [44]. The leaf sections of all 435 individuals were double-stained with safranin (1 g of safranin dissolved in 100 mL of 50% ethanol) and alcian blue (1 g of dye dissolved in 100 mL of distilled water, with a couple of crystals of phenol and three drops of glacial acetic acid). Stained sections were then dehydrated through an alcohol series (50%, 70%, 96%, 100%), examined, and photographed with a Leica DM 1000 microscope (Leica Microsystems) [16,45,46]. Anatomical features were measured in ImageJ [47]. A list of 42 characters from the literature relevant for this group [46,48,49] related to morphology and leaf anatomy was taken into consideration (see Appendix A). The qualitative characters were standardized as states represented by numbers (see Appendix A).

Principal component (PCA) and discriminant (CDA) analyses for morpho-anatomical characters were computed using Statistica 7.0 [50]. Due to unbalanced sample numbers (two populations of diploid vs. 13 populations of polyploid *C.* cf. *heuffelianus*), the set of differential characters was defined based on representative populations for each taxon/cytotype derived from type localities (140 individuals; Appendix A), computing a PCA. PCA served as a tool to point out the significant traits. The characters highlighted as important with the PCA were furthermore used for CDA. Moreover, eight a priori-defined groups of parental species and polyploids were included in CDA: *C. vernus* (100 individuals); *C. heuffelianus* (40); mixed *C. heuffelianus*/*C.* cf. *heuffelianus* 2*n =* 18 SCC (40); *C.* cf. *heuffelianus* 2*n =* 18 SCC (60); *C.* cf. *heuffelianus* 2*n =* 18 WCC (35); *C.* cf. *heuffelianus* 2*n =* 18 PIC (40); *C.* cf. *heuffelianus* 2*n* = 20 (80); *C.* cf. *heuffelianus* 2*n =* 22 (40) (435 individuals; Appendix A). The CDA results were plotted with ggplot2 [34].

## 3. Results

### 3.1. Determination of Ploidy Levels

**Chromosome counts**—Most chromosome counts obtained in this study coincided with previous reports (Appendix A). The lowest chromosome count was in *C. etruscus*, *C. ilvensis*, and *C. vernus* (2*n =* 8) followed by *C. heuffelianus* (2*n =* 10); *C. bertiscensis* (2*n =* 12); *C. kosaninii* (2*n =* 14); *C. neglectus*, *C. tommasinianus*, and *C.* cf. *vernus* (2*n =* 16); PIC, SCC, and WCC populations of *C.* cf. *heuffelianus* (2*n =* 18); *C.* cf. *heuffelianus* from Montenegro and Serbia (2*n =* 20); *C.* cf. *heuffelianus* from Albania and Kosovo (2*n =* 22); and *C. longiflorus* (2*n =* 28). The chromosome count of a *C. vernus*-derived polyploid population from Central Albania, herein referred to as *C.* cf. *vernus* (2*n* = 16), is reported here for the first time (Appendix A).

**Genome sizes**—The smallest genome size was observed in *C. longiflorus* (2C = 3.21 pg), followed by *C. tommasinianus* (2C = 5.53 pg) and *C. vernus* (2C = 5.78 pg). *Crocus bertiscensis* was observed with an average genome size of 2C = 6.66 pg. *Crocus etruscus* (2C = 7.58 pg), *C. ilvensis* (2C = 7.88 pg), *C. heuffelianus* (2C = 7.73 pg), and *C. kosaninii* (2C = 7.95 pg) had similar genome sizes (Appendix A). Populations of *C. heuffelianus* with higher chromosome counts (2*n* = 18, 20, 22) were measured with average genome sizes per cytotype ranging between 2C = 10.88 and 2C = 12.84 pg (Table 1). Similar genome sizes were measured for *C. neglectus* (2C = 12.24 pg) and *C.* cf. *vernus* (2C = 12.23 pg).

Plotting of genome sizes against the chromosome counts showed a general negative relationship between genome size and chromosome number in both diploid and tetraploid taxa. This relationship is weaker when *C. longiflorus* is excluded (Appendix A). *Crocus longiflorus* is sister to the core series *Verni* taxa and has a seemingly polyploid chromosome count (2*n* = 28), although genome size indicates a diploid genome (Appendix A).

**GBS-derived heterozygosity to ploidy**—Most SNPs in the datasets were bi-allelic even in polyploids. The heterozygosity H_0_ ranged between 0.0115 and 0.0628 for the GBS data set used (Appendix A). Individuals can be divided into two groups: samples with a H_0_ of 0.0115 to 0.0278 and samples with a H_0_ of 0.0359 to 0.0628 (Appendix A). If heterozygosity is considered in the context of known genome sizes, the group with the higher H_0_ possesses the larger genome sizes (2C = >10 pg). The group with lower H_0_ also has smaller genome sizes (2C = <8 pg) and generally lower chromosome numbers (except for *C. longiflorus*) and comprises *C. vernus* (2C = 5.78 pg, 2*n =* 8), *C. etruscus* (2C = 7.58 pg, 2*n =* 8), *C. ilvensis* (2C = 7.88 pg, 2*n =* 8), *C. heuffelianus* (2C = 7.73 pg, 2*n =* 10), *C. bertiscensis* (2C = 6.66 pg, 2*n =* 12), *C. kosaninii* (2C = 7.95 pg, 2*n =* 14), *C. tommasinianus* (2C = 5.53 pg, 2*n =* 16), and *C. longiflorus* (2C = 3.21 pg, 2*n =* 28). As a consequence, all samples with a H_0_ below 0.03 are considered to represent diploids, while all samples with a H_0_ above 0.035 are considered polyploids. The latter group includes *C.* cf. *heuffelianus* with higher chromosome counts and genome sizes (2C = 10.88 to 12.84 pg, 2*n =* 18, 20, 22), *C. neglectus* (2C = 12.24 pg, 2*n =* 16), and *C.* cf. *vernus* (2C = 12.23 pg, 2*n =* 16).

### 3.2. Phylogenetic Inference and Origin of Allopolyploids

**GBS-derived data**—Initially, we created a single dataset for all GBS-derived analyses in ipyrad, i.e., including all 194 sequences in one alignment (2009 loci, 187,846 bp alignment length, 20.88% missing sites). From this, we derived the dataset that includes only the diploid accessions by excluding all polyploid individuals from the data matrix (Table 2). The SVDquartets analysis of the diploid species (Figure 1) using the multispecies coalescent was used to infer the phylogenetic relationships among the diploid species, which are the basic taxonomic units in this group. Here *C. longiflorus* (2*n* = 2*x* = 28) from Sicily is the sister species to all other taxa within series *Verni*. The next two consecutively branching clades consist of the species with 2*n* = 2*x* = 8 with *C. ilvensis* grouping together with *C. etruscus* followed by the clade of *C. neapolitanus*, *C. siculus,* and *C. vernus*. These species all occur in Italy or the Alps. The latter group is sister to a clade that harbors the eastern species *C. heuffelianus*, *C. bertiscensis*, *C. kosaninii,* and *C. tommasinianus* with the higher chromosome numbers of 2*n* = 2*x* = 10, 12, 14, and 16, respectively.

The MP analysis of diploid taxa (Figure 2, Appendix A) differs strongly from the diploid’s SVDquartets tree topology. *Crocus longiflorus* is in both analyses sister to all other series *Verni* species. In the next clade the positions of the 2n = 8 and 2n ≥ 10 taxa are, however, reversed. Although the species in the tree received very high bootstrap support, for the clades along the backbone of the tree, support values are low (Appendix A), so that the topology of this tree has no strong support. We used MP mainly to infer the topology of the polyploids in relation to their diploid progenitors. In MP allopolyploids mostly group within or as sister to the parental species where they share higher genetic similarity and, as MP is sensitive to reticulate data structure, indicates if a polyploid is monophyletic or might consist of different subgroups. In the analysis of the combined di- and tetraploid GBS data (Figure 3, Appendix A), *C. longiflorus*, *C. kosaninii*, and *C. bertiscensis* are the first species branching off, with the latter being sister to a large clade consisting of three subclades: (a) *C. etruscus*, *C. ilvensis,* and *C. neglectus* being sister to *C. tommasinianus*, (b) *C. neapolitanus*, *C. siculus,* and *C. vernus* as sister group of (c) *C. heuffelianus*. The polyploids were grouped in between the diploid taxa (Figure 3, Appendix A).

**Chloroplast-derived data**—Series *Verni* diploids are split into two major groups in the phylogenetic tree of combined chloroplast data. The first group comprises *C. heuffelianus* (2*n* = 2*x* = 10), *C. bertiscensis* (2*n* = 2*x* = 12), *C. kosaninii* (2*n* = 2*x* = 14), and *C. tommasinianus* (2*n* = 2*x* = 16) (Figure 4, Appendix A). The second group consists of two subgroups: (a) a group comprising the 2*n* = 2*x* = 8 species *C. etruscus, C. ilvensis, C. neapolitanus*, *C. siculus,* and *C. vernus* from its northern and western distribution range (Alps to Pyrenees, NW type), and (b) *C. vernus* from the southeastern distribution range (Dinaric Alps; SE type), which has *C. longiflorus* (2*n* = 2*x* = 28) as its sister group.

The allotetraploids were grouped with their maternal parents. *Crocus neglectus* (2*n =* 4*x* = 16) possesses a chloroplast similar to *C. ilvensis*. *Crocus* cf. *heuffelianus* (WCC; 2*n* = 4*x* = 18) from Slovakia groups with *C. heuffelianus*. Most of the *C.* cf. *heuffelianus* (PIC; 2*n* = 4*x* = 18) individuals from Bosnia and Herzegovina as well as Slovenia grouped in a clade with the western 2*n* = 2*x* = 8 species and are partly identical with the NW chloroplast type of *C. vernus* (e.g., *C. vernus* from Slovenian Alps is identical with *C.* cf. *heuffelianus* from Bosnia and Herzegovina). However, we also found *C.* cf. *heuffelianus* PIC grouping with the SE chloroplast type of *C. vernus,* indicating an independent origin. *Crocus* cf. *heuffelianus* (SCC; 2*n* = 4*x* = 18) from Romania, 2*n* = 4*x* = 20 from Montenegro and Serbia, 2*n* = 4*x* = 22 from Northern Albania and Kosovo, as well as *C*. cf. *vernus* (2*n* = 4*x* = 16) from Central Albania were found in the clade with the southeastern *C. vernus* populations (SE type).

**Nuclear single-copy markers**—Three variable nuclear single-copy regions (*orcp*, *rcf*2, *topo*6; Appendix A) were chosen to identify the position of alleles of the alloploids by the criterion that the marker regions showed differences between the potential diploid parental species. Up to four alleles could be found within the allopolyploids (Figure 5, Appendix A). *Crocus* cf. *heuffelianus* 2*n* = 4*x* = 18 (WCC) and *C.* cf. *heuffelianus* 2*n* = 4*x* = 18 (PIC) were found to be grouping with *C. vernus* from its western distribution range and/or with *C. heuffelianus*. *Crocus* cf. *heuffelianus* 2*n* = 4*x* = 18 (SCC) shared similar alleles with *C. vernus* from its eastern distribution range and/or *C. heuffelianus*. The same applies for *C.* cf. *heuffelianus* 2*n* = 4*x* = 20 and 2*n* = 4*x* = 22. *Crocus neglectus* (2*n* = 4*x* = 16) grouped with *C. ilvensis* or within a clade comprising *C. neapolitanus* as well as other allotetraploids in two of the selected nuclear single-copy genes. Generally, the gene-tree topologies were different from the topologies in chloroplast or GBS-derived datasets and differences also occurred among the single-copy genes. For example, in *topo*6, *C. longiflorus* occupies a similar position as in the chloroplast marker tree where it groups in one clade with other Italian taxa such as *C. neapolitanus*, *C. ilvensis,* and *C. vernus* (Figure 5). In the *rcf*2-derived tree its position is similar to the GBS results, where it groups as sister to the core series *Verni* taxa. *Crocus tommasinianus*, or one of its alleles, was found in one clade with *C. vernus* (NW) in *topo*6 and *rcf*2 (Figure 5), while it groups with *C. bertiscensis*, *C. kosaninii,* and *C. heuffelianus* in the SVDquartets of GBS and the chloroplast trees. The *orcp* data set had the highest number of parsimony-informative sites of the three closely examined nuclear single-copy genes, but also the highest homoplasy and was mostly characterized by polytomies (Table 2, Appendix A).

**ITS**—In the ITS tree (Appendix A) *C. longiflorus* is sister to the other series *Verni* taxa (pp 1.0). In the sister clade of *C. longiflorus*, *C. kosaninii* is separated from the remaining species of the series but with very low support (pp 0.61). The majority of series *Verni* species is found in a large polytomy with only a few subclades. In one of the subclades, *C. ilvensis* groups with *C. etruscus*, while most of the other subclades are formed by samples of the same species, with the exception of the subclade comprising *C. vernus*. Here, *C. neapolitanus*, *C. siculus*, *C. vernus*, and all allotetraploids included in the dataset can be found (pp 1.0; Appendix A), albeit their relationships remained unresolved.

### 3.3. Phylogenomic Analysis

The 192 GBS sequences of series *Verni* (excluding the outgroup *C. malyi*) with a threshold number of 120 samples sharing a locus resulted in a dataset comprising 2207 loci with 18.38% missing sites. The data matrix of unlinked SNPs included 2172 SNPs. The sample with the lowest number of reads (559,922) and loci (866) was *C. siculus*. 

The GBS-based PCA (Figure 6) placed *C. tommasinianus* clearly separate from all other samples in the negative part of the PC1 axis. In between *C. tommasinianus* and the majority of all other taxa, in the positive part of the PC1, *C. heuffelianus* and *C. vernus* can be found. However, *C. heuffelianus* in the positive part of the PC2 axis (5 or higher) and *C. vernus* in the negative part of the PC2 axis (−4 or lower) were distinct from each other. The two individuals of *C. neapolitanus* were close to *C. vernus*. *Crocus etruscus* and *C. ilvensis* were placed together in close proximity to *C. bertiscensis*, *C. kosaninii*, and *C. longiflorus* representatives, all between −1 and 1 on the PC1 axis and 0 to 4 on the PC2 axis. *Crocus siculus* was found in the lower negative part of the PC2 axis (ca. −2), partly overlapping with the polyploids *C.* cf. *heuffelianus*, *C. neglectus,* and *C.* cf. *vernus*. They were placed in between *C. heuffelianus* and *C. vernus* but always in the positive part of the PC1 axis.

Co-ancestry analysis with RADpainter showed admixture for *C. neglectus* (Appendix A) with both *C. etruscus* (co-ancestry 234.2, 241.8) and *C. ilvensis* (co-ancestry 217.9, 221.8) as well as with *C. neapolitanus* (co-ancestry 99.2, 103.7). *Crocus etruscus* cv. ‘Zwanenburg’ shares a relatively high co-ancestry with *C. etruscus* (380.6, 386.3) followed by *C. ilvensis* (301.3, 301.1) and *C. tommasinianus* from Italy (110.5). *Crocus* cf. *tommasinianus* was found to be admixed with *C. vernus* (181.7) and *C. tommasinianus* (135.1) growing at the same location (Montenegro, Mt. Lovcen). The highest co-ancestry of *C.* cf. *heuffelianus* 2*n* = 4*x* = 18 (PIC) and diploid series *Verni* species was found with *C. vernus* (52.8–86.9), while it was lower with *C. heuffelianus* (44.0–51.6). The level of co-ancestry for *C.* cf. *heuffelianus* 2*n* = 4*x* = 18 (SCC) was high in the mixed-ploidy populations with *C. heuffelianus* (127.5–128.8), while it was between 50 and 101.5 with other *C. heuffelianus* populations and between 49.2 and 59.4 with *C. vernus*. *Crocus* cf. *heuffelianus* 2*n* = 4*x* = 18 from the Western Carpathians (WCC) had its highest co-ancestry level with *C. heuffelianus* (49.4–93.4). Its co-ancestry shared with *C. vernus* ranged between 51.6 and 62.0. The co-ancestry levels of *C.* cf. *heuffelianus* 2*n* = 4*x* = 20 and 2*n* = 4*x* = 22 shared with *C. vernus* were lower than in the other allotetraploid *C.* cf. *heuffelianus* (48.0–58.2 and 47.8–58.5). The same applies to the shared co-ancestry with *C. heuffelianus* (46.1–54.0 and 45.9–52.2). In addition, they are even lower than the co-ancestry shared between diploid species such as *C. bertiscenesis* and *C. kosaninii* (57.1–62.9).

In the population assignment analysis (K with the lowest entropy was K = 8), most of the alleles present in *C. tommasinianus* samples were assigned to one ancestral population (Appendix A). Moreover, *C. vernus* alleles were mostly assigned to one ancestral population, and representatives of *C. heuffelianus* were mostly assigned to their own ancestral population as well. However, *C. bertiscensis*, *C. etruscus*, *C. ilvensis*, *C. kosaninii,* and *C. longiflorus* appeared admixed, partly sharing *C. heuffelianus* and *C. vernus* (*C. etruscus*, *C. ilvensis,* and *C. longiflorus*) patterns or additionally having alleles assigned to *C. tommasinianus* and/or to *C.* cf. *heuffelianus*. The different *C.* cf. *heuffelianus* groups were partly assigned to their own ancestral population showing no admixture with LEA (K = 5, K = 8) or to *C. vernus* (K = 4) with fastStructure. *Crocus* cf. *vernus* and *C. neglectus* were partly assigned to the *C. vernus* ancestral population but also to different ancestral populations of *C.* cf. *heuffelianus*. *Crocus etruscus* cv. ‘Zwanenburg’ and *C.* cf. *tommasinianus* showed admixture, with parts of their alleles derived from *C. tommasinianus*. In the case of *C.* cf. *tommasinianus*, *C. vernus* contributed genomic materials, and *C. etruscus* cv. ‘Zwanenburg’ was complemented by *C. etruscus*.

Assigning the several polyploid groups mostly to one ancestral population while showing diploids as admixed was also observed for lower K or other SNP subsampling and/or other ancestral assignment methods (Appendix A) regardless of the analysis program used.

*Crocus* cf. *heuffelianus* had the lowest F*st* with *C. heuffelianus* and *C. vernus* (2*n* = 4*x* = 18 WCC: F*st* = 0.24 and F*st* = 0.22; 2*n* = 4*x* = 18 SCC: F*st* = 0.14 and F*st* = 0.17; 2*n* = 4*x* = 18 PIC: F*st* = 0.17 and F*st* = 0.12; 2*n* = 4*x* = 20: F*st* = 0.18 and F*st* = 0.15; 2*n* = 4*x* = 22: F*st* = 0.22 and F*st* = 0.19) or, in cases of *C.* cf. *heuffelianus* (2*n* = 4x = 18 PIC), with *C. neapolitanus* (F*st* = 0.15; Appendix A). The lowest F*st* for *C.* cf. *vernus* was observed towards *C. vernus* (F*st* = 0.15), followed by *C. neapolitanus* (F*st* = 0.20) and *C. heuffelianus* (F*st* = 0.23). *Crocus neglectus* had the lowest F*st* towards *C. etruscus* (F*st* = 0.07), *C. vernus* (F*st* = 0.08), *C. neapolitanus* (F*st* = 0.10), and *C. ilvensis* (F*st* = 0.10). The cultivar ‘Zwanenburg’ had the lowest F*st* (0.00) with *C. etruscus*, followed by *C. ilvensis* (F*st* = 0.09) and *C. tommasinianus* (F*st* = 0.14). The lowest F*st* for *C.* cf. *tommasinianus* was found with *C. tommasinianus* (F*st* = 0.14) and *C. vernus* (F*st* = 0.13).

### 3.4. Morpho-Anatomical Analyses

The PCA of the morpho-anatomical dataset highlighted 14 characters with PC scores >0.70: outer and inner perigone segment length and width (Outer_ps_l, Outer_ps_w, Inner_ps_l, Inner_ps_w), anther length (Anther_l), throat hair (Th_hair), leaf section height and width (Sectioh_h, Section_w), arm length (Arm_l), central parenchyma area (Parenchyma_a), palisade cell and tissue height (Pal_cell_h, Pal_tissue_h), spongy tissue height (Sp_tissue_h), and xylem area (Xy_a) (Appendix A). This set was extended with stigma/anther ratio (S/a_r) (PC1 = 0.68) as the most important discriminative feature for *C. vernus* confirmed by previous research [15]. Finally, the CDA of the complete dataset (435 individuals of *C. vernus*, *C. heuffelianus*, mixed *C. heuffelianus*/*C.* cf. *heuffelianus* 2*n =* 18 SCC, *C.* cf. *heuffelianus* 2*n =* 18 SCC, *C.* cf. *heuffelianus* 2*n =* 18 WCC, *C.* cf. *heuffelianus* 2*n =* 18 PIC, *C.* cf. *heuffelianus* 2*n =* 20, and *C.* cf. *heuffelianus* 2*n =* 22) was computed based on 15 previously mentioned characters. The clear separation of *C. heuffelianus* (Figure 6) in the negative part of the CDA of both axes was caused by the absence of throat hair (Figure 7, Appendix A). The mixed populations of the diploid *C. heuffelianus* and its polyploid 2*n* = 18 SCC cytotype overlapped with these two taxa (Figure 6), while all other polyploid populations were grouped in the positive part of both axes (Figure 6). The characters responsible for differentiation along the second axis were leaf cross-section width and arm length (Appendix A).

## 4. Discussion

### 4.1. Recognition of Recent Polyploids and Their Parents

*Crocus heuffelianus* represents one of the biggest taxonomical challenges within *Crocus* due to its high morphological variability. This morphological diversity seems mainly caused by the allotetraploid origin of the karyotypes of taxa possessing higher chromosome counts (2*n* = 18, 20, 22), which became evident in a former molecular study [15]. Through incongruences between the GBS tree (Figure 1, Figure 2 and Figure 3) and the chloroplast tree (Figure 4) as well as to the ITS tree (Appendix A) and/or the affiliation of different alleles of variable nuclear single-copy markers (Figure 5, Appendix A) we identified at least seven independent hybridization events involving *C. heuffelianus* (2*n* = 10) and *C. vernus* (2*n* = 8), mostly with *C. vernus* as maternal parent. We summarize these reticulate relationships in Figure 8, a tree based on the diploid’s topology derived from the SVDquartets analysis. *Crocus vernus* was found to possess two different chloroplast types depending on the geographical distribution (Eastern Alps to Pyrenees: NW type; Dinaric Alps: SE type). *Crocus vernus* from the Alps hybridized *C. heuffelianus* resulting in allotetraploid Western Carpathian populations (WCC; 2*n* = 18) with *C. heuffelianus* as the maternal parent. In the Pannonian-Illyric group of *C.* cf. *heuffelianus* (PIC; 2*n* = 18), we found two different chloroplast haplotypes stemming from the SE and NW *C. vernus* types indicating two crosses involving *C. vernus* as maternal parents (Figure 8). Furthermore, we also observed differences among the chloroplast haplotypes stemming from the NW type of *C. vernus* (Figure 4). One differed by at least two substitutions and two indels from any other NW type (see also branch lengths in Appendix A). This indicates multiple hybridization events between *C. vernus* as maternal and *C. heuffelianus* as paternal parent creating the Pannonian-Illyric *C.* cf. *heuffelianus*. The SE type-carrying *C. vernus* was identified as the maternal species for Southern Carpathian populations of *C.* cf. *heuffelianus* (SCC; 2*n* = 18), as well as for the cytotypes 2*n* = 4*x* = 20 and 2*n* = 4*x* = 22 (Figure 8). 

Considering that the Southern Carpathian diploid cytotype of *C. heuffelianus* (2*n* = 10) is ancestral to all polyploid forms and that according to morphological and chorological characteristics it corresponds to the original description, it represents *C. heuffelianus* s. str. [16,49]. It comprises populations with darker perigones and predominantly glabrous throats or hardly visible sparse and short hairs, which makes it distinct from all other series *Verni* species. Several authors reported different distribution ranges for *C. heuffelianus* [51,52,53]. This confusion is likely caused by confusing *C. heuffelianus* s. str. with its morphologically similar allopolyploids. For instance, some authors reported *C. heuffelianus* s. str. to occur in regions that are, according to our investigations, only inhabited by allopolyploid *C.* cf. *heuffelianus*. (e.g., throughout Bosnia and Herzegovina [54] or northeastern Italy [55]).

As a result of our study, we confirm a Carpathian distribution of *C. heuffelianus* s. str. in Slovakia, Romania, and Ukraine. *Crocus* cf. *heuffelianus* allopolyploid cytotypes have partly sympatric distributions with one of their parents or are growing in between the parental distribution areas (Appendix A).

A new polyploid, *C.* cf. *vernus* (2*n* = 4*x* = 16), was found in Central Albania, having *C. vernus* as the maternal parent. Its hybrid origin is indicated by its sister position in the GBS phylogeny, where it did not group within *C. vernus* as it would if it were an autopolyploid [5]. A (segmental) allotetraploid origin is also evident by its position in the SNP-based PCA (Figure 6). In the three closely examined variable nuclear single-copy genes (Figure 5, Appendix A), some of *C.* cf. *vernus*’s alleles were unique or usually grouped close to those of other taxa with 2*n* = 8 chromosomes. Therefore, the paternal parent could either be an extinct *C. vernus*-like genotype, which probably had eight chromosomes, or stems from a *C. vernus* population that we have not yet collected.

Allopolyploids in *C.* cf. *heuffelianus* and *C.* cf. *vernus* are also genetically differentiated, as indicated by the F*st* values, which were usually higher than 0.15. This genetic differentiation might explain the difficulty in assigning ancestral populations (Appendix A) or in the inference of co-ancestry (Appendix A), where some allotetraploids were not shown as admixed.

*Crocus neglectus* (2*n* = 4*x* = 16) could be confirmed here as an allotetraploid with *C. ilvensis* or *C. etruscus* as the maternal parent [15]. The fixation index (*C. etruscus* F*st* = 0.07; *C. ilvensis* F*st* = 0.10) points to a more likely contribution of *C. etruscus*. However, *C. neglectus* shares its chloroplast haplotype with *C. ilvensis*, which was not found in *C. etruscus*. This discrepancy may be explained by the possibility that genetic drift eliminated this type of chloroplast in *C. etruscus* while it persisted in the geographically isolated *C. ilvensis* or that it was just not discovered in the individuals analyzed up to now. Seed size and germination are also more similar to *C. etruscus* [56], while flower bouquet is more similar to *C. ilvensis* [57]. *Crocus neapolitanus* is likely the paternal parent of *C. neglectus*. *Crocus neapolitanus* is genetically very similar to *C. vernus* but has a species-specific allele in one of the nuclear single-copy markers (*rcf*2; Figure 5) found in *C. neglectus*. The relatively high co-ancestry shared between *C. neglectus* and *C. neapolitanus* further supports *C. neapolitanus* as paternal parent. 

### 4.2. General Results Regarding Phylogeny and Systematics

Until the advent of high-throughput sequencing technologies, phylogenetic studies were often restricted to a limited set of markers. Consequently, relationships could not be resolved, as in the case of series *Verni* [15]. In Raca et al. [16], we already successfully increased the resolution by using genome-wide SNP data obtained from GBS. However, since this study aimed to show the phylogenetic affiliation of *C. bertiscensis*, we included neither all the species in *Crocus* ser. *Verni* (*C. siculus* was lacking) nor a higher number of individuals per species and did not analyze species relationships in detail. Here we added additional samples, a chloroplast marker dataset, as well as a dataset comprising nuclear single-copy genes. The latter mainly served to identify the parental origin of the recent allotetraploids. Excluding the allotetraploids in the analysis increased the support values of the tree backbone of the GBS-based MP tree, similar to the BI-based analysis in Raca et al. [16]. However, our SVDquartets species tree (Figure 1) showed a different topology. The relatively high degree of homoplasy in the GBS dataset (Table 2) indicates incomplete lineage sorting (ILS) and/or hybridization that could result in genomic introgression. Incongruences of single-gene trees, as well as the chloroplast tree, support this hypothesis. For example, *C. vernus* today has two different chloroplast haplotypes (SE type and NW type). The NW type is shared with other closely related species such as *C. etruscus*, *C. ilvensis*, *C. neapolitanus*, and *C. siculus*, which all occur in Italy (Appendix A). The SE type is found only in the southeastern range of *C. vernus* and groups with *C. longiflorus* as sister to the NW type clade. A possible interpretation could be that these two chloroplast types were once both present in Italy, where introgression of *C. vernus* or its ancestor with *C. longiflorus* occurred. Subsequently, the SE type was sorted out and persisted only in the eastern distribution of *C. vernus*. The position of *C. longiflorus* as sister to *C. vernus* was also observed in one of the nuclear single-copy markers (*topo*6), which supports the hypothesis of an ancient introgression event between *C. vernus* or its ancestor and *C. longiflorus*. There are also other examples where species (or one of their alleles) group with species to which they are only distantly related according to the GBS trees (SVDquartets as well as MP), such as *C. tommasinianus* grouping together with *C. vernus* in *topo*6 and *rcf*2 (Figure 5).

Single gene trees generally showed a poor resolution, which was the main reason we show only *rcf*2 and *topo*6, and even in data sets with a relatively high number of parsimony-informative sites, phylogenetic relationships remained largely unresolved (Appendix A) due to the young age of the group and homoplasy in the data. Since multispecies coalescent methods consider ILS and introgression, the SVDquartets tree should more closely reflect the true species relationships although its topology differs from trees derived with concatenation approaches [58].

### 4.3. Chromosome and Genome Size Evolution

All angiosperms essentially have undergone polyploidization [59]. Dysploidy, chromatin elimination or expansion, nested polyploidizations, introgression, and hybridization can confound the evolutionary dynamics between chromosome number and genome size over time [60,61,62]. Consequently, these two genomic parameters show independent evolution and no clear correlation, especially in older polyploids, such as meso- and paleo-polyploids [61,63,64]. Nevertheless, some correlation can still be observed in neopolyploids [6,61].

Neopolyploids often have roughly additive genome sizes and chromosome numbers of the progenitors, as in *C.* cf. *heuffelianus* (2*n* = 18, 2C = 10.88–12.84 pg), *C. neglectus* (2*n* = 16, 2C = 12.24 pg), and *C.* cf. *vernus* (2*n* = 16, 2C = 12.38 pg). However, although genome sizes could remain roughly additive in the absence of considerable chromatin loss [65], the additive pattern in chromosome number is often blurred when chromosome fusion (descending dysploidy) or fission (ascending dysploidy) occurs [61,66,67]. Considering the chromosome numbers and genome sizes of the parental genomes of *C.* cf. *heuffelianus*, *C. heuffelianus* (2*n* = 10), and *C. vernus* (2*n* = 8), it is highly likely that the *C.* cf. *heuffelianus* cytotypes from Albania, Kosovo, Montenegro, and Serbia (2*n* = 20 and 2*n* = 22, 2C = 11.95 pg) have undergone ascending dysploidy. Indeed, a few shorter chromosomes have been observed in these cytotypes, as ascending dysploidy entails chromosome breakage resulting in an increase in chromosome number in general and shorter chromosomes in particular (Appendix A).

The cryptic relationship between chromosome number and genome size increases over time in polyploids. This relationship can be observed between *C. longiflorus* and the core series *Verni* species. In general, series *Verni* species with more chromosomes have smaller genomes, thus showing a negative relationship between genome size and chromosome number, with or without *C. longiflorus* (Appendix A), which is sister to all other taxa of series *Verni* and was recently moved to this series from the now-defunct series *Longiflori* B.Mathew [15]. *Crocus longiflorus* has the highest chromosome number but the lowest genome size (2*n* = 28, 2C = 3.21 pg) in the series. Likewise, *C. tommasinianus* (2*n* = 16), which has twice as many chromosomes as *C. vernus* (2*n* = 8), has a roughly similar genome size to *C. vernus*. Taking into account only chromosome counts, *C. longiflorus* and *C. tommasinianus* would likely be deemed polyploids. However, they show very low heterozygosity scores (Appendix A), indicating extensive chromatin elimination and essentially diploidized genomes. This chromosome number–genome size relationship between *C. longiflorus* and the core species of series *Verni* can indicate an ancient whole-genome duplication event prior to the divergence of series *Verni*, likely even before the diversification of *Crocus* [68].

A burst of certain repetitive DNA element families may also have promoted genome size expansion in the core series *Verni*. Thus, the genome sizes in the core taxa of series *Verni* have increased despite chromosome number reduction relative to *C. longiflorus*. Nevertheless, this hypothesis can only be supported when fused chromosome blocks [59,69] and expansion of lineage-specific repeat families [70,71] are observed within the core of series *Verni*. This can become possible by comparative genome and repeat analyses between *C. longiflorus* and the other species from series *Verni*. Combined with cytogenetic analyses involving all major groups within the genus, this should allow an understanding of karyotype evolution within the genus.

## 5. Conclusions

Our study was designed to disentangle the *C. heuffelianus* complex using GBS data and chloroplast markers and combine them with morphology, genome sizes, and chromosome counts. This strategy generally proved to be successful when (a) a broad sampling of allotetraploids and potential parental species are included and (b) the allotetraploids group as sister to their paternal parent in the phylogenetic GBS tree. In cases where our sampling was restricted to just a few samples, such as for some of the Italian species and allotetraploid *C. neglectus*, conclusions remain a bit uncertain. The combination of chloroplast data with only GBS failed to reveal the paternal contributor of *C.* cf. *heuffelianus* (SCC; 2*n* = 18). Here, additional nuclear markers (Figure 5, Appendix A) were necessary to identify the NW type of *C. vernus* as the paternal parent. Algorithms specifically designed to detect hybridization signals in GBS data were only partly able to recover the allopolyploids within series *Verni*, while in several cases co-ancestry values for their parents were lower than even between independent diploids. Thus, such an approach alone is not sufficient to infer polyploidization or indicate the diploid progenitors of polyploids in crocuses.

By linking molecular results and genome sizes with morphology, a clear differentiation of allopolyploids and parental species was possible. In the taxonomically confusing *C. heuffelianus* complex, a circumscription of the diploid taxon and its distinction from the allotetraploids is now possible. *Crocus heuffelianus* s. str. is the diploid cytotype (2*n* = 10) with mostly glabrous throats and darker perigone segments. Together with *C. vernus*, it represents the parental species for all the *C.* cf. *heuffelianus* allotetraploids. The cytotype 2*n* = 18 of *C.* cf. *heuffelianus* is split into three groups: Western Carpathian (WCC), Pannonian-Illyric (PIC), and Southern Carpathian (SCC). *Crocus heuffelianus* s. str. is the mother of WCC only, while the NW and SE types of *C. vernus* are maternal lineages of PIC. The SE type of *C. vernus* is the only maternal parent of SCC, as well as for the cytotypes with 2*n* = 20 and 2*n* = 22 chromosomes. All analyzed *C.* cf. *heuffelianus* polyploids represent morphologically intermediate forms between their parental species, but currently, they cannot be distinguished based on the investigated morphological characters. Instead, chromosome counts are necessary.

While it is possible to unravel more recent polyploidization events, the detection of paleo-polyploidization remains difficult. Our incongruent gene trees indicate past hybridization events, which might have triggered genome size and chromosome number changes. However, while the methods applied here work well in this very young taxon group, they are not satisfactory in uncovering ancient and complex evolutionary histories, particularly those involving highly dynamic genome size and chromosome number changes. In series *Verni*, and in *Crocus* in general, the future availability of genome assemblies will enable comparative cytogenomic analyses to detect potential ancient polyploidization and to trace chromosomal rearrangements resulting in changing karyotypes.

## Figures and Tables

**Figure 1 biology-12-00303-f001:**
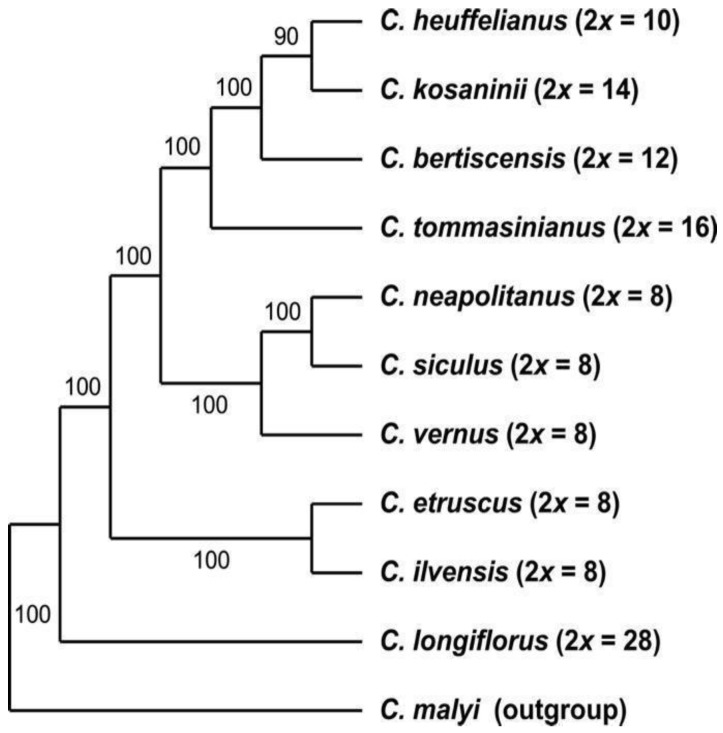
Phylogenetic tree of diploid species and accessions of *Crocus* ser. *Verni* based on the GBS dataset analyzed with SVDquartets analysis using the multispecies coalescent as tree model. Numbers along branches indicate bootstrap support values; chromosome numbers are provided in brackets.

**Figure 2 biology-12-00303-f002:**
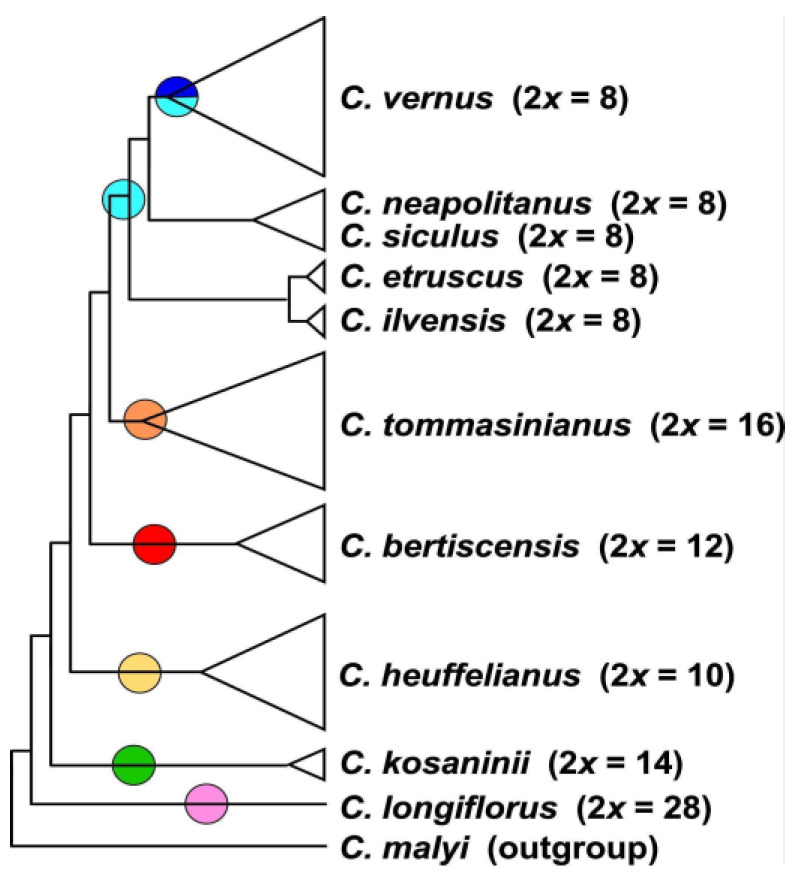
Schematic representation of the strict consensus tree topology of 45 most parsimonious trees derived from an MP analysis of the GBS dataset including only diploid accessions of *Crocus* ser. *Verni* (for original data see Appendix A). In brackets, the ploidy levels and chromosome numbers are given. Colored circles refer to the chloroplast types present in the respective clades (below).

**Figure 3 biology-12-00303-f003:**
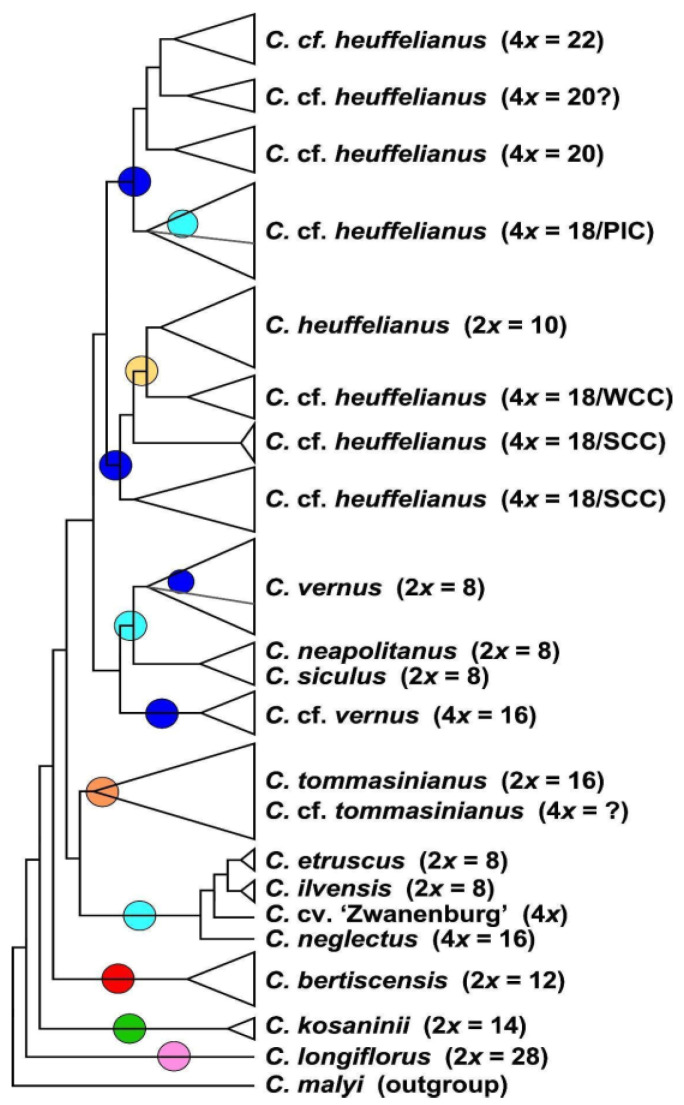
Schematic representation of the strict consensus tree topology of 5400 most parsimonious trees derived from an MP analysis of the GBS dataset including di- and tetraploid accessions of *Crocus* ser. *Verni* (for original data see Appendix A). In brackets the ploidy levels and chromosome numbers are given. For tetraploids with 4*x* = 18 the geographic affiliation is also provided, as Pannonian-Illyric Clade (PIC), Southern Carpathian Clade (SCC), and Western Carpathian Clade (WCC). Colored circles refer to the chloroplast types present in the respective clades (below).

**Figure 4 biology-12-00303-f004:**
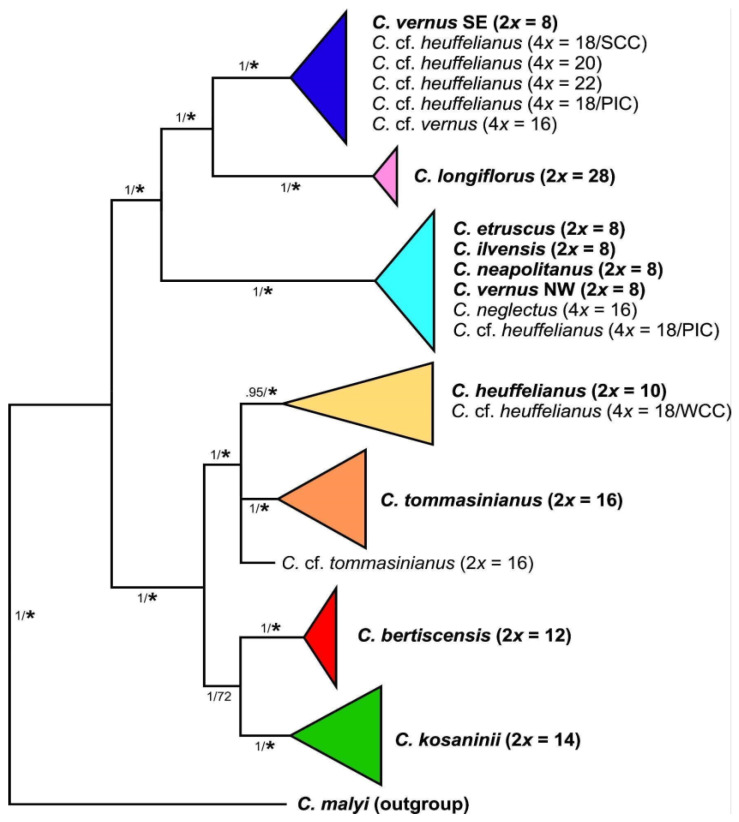
Schematic representation of relationships of species and cytotypes of *Crocus* ser. *Verni* obtained through Bayesian phylogenetic inference of sequences from two chloroplast regions (for original data see Appendix A). Diploid accessions, as basic units in the series, are given in boldface. Numbers along branches indicate Bayesian posterior probabilities/maximum parsimony bootstrap values (≥50%), with asterisks for bootstrap support values >80%.

**Figure 5 biology-12-00303-f005:**
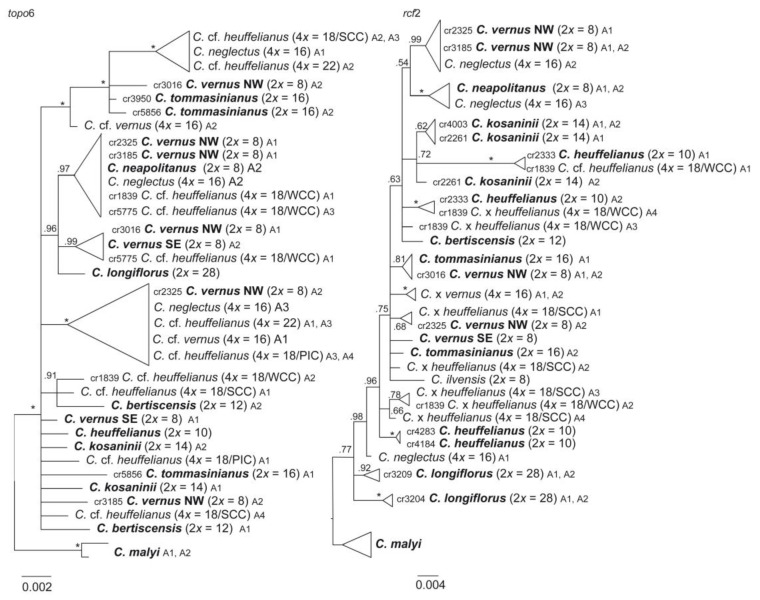
Phylogenetic trees obtained through Bayesian phylogenetic inference based on haplotype sequences of nuclear single-copy genes in *Crocus* ser. *Verni*. Numbers along branches indicate BI posterior probabilities (pp). Support values of 1.0 are indicated by asterisks. Allelic differences (A1–A4) in these markers were used to track the bi-parental contributions of diploids to allotetraploids. If more than one individual per species or cytotype was included, their DNA identifier (cr number) is provided. For detailed information, see Appendix A.

**Figure 6 biology-12-00303-f006:**
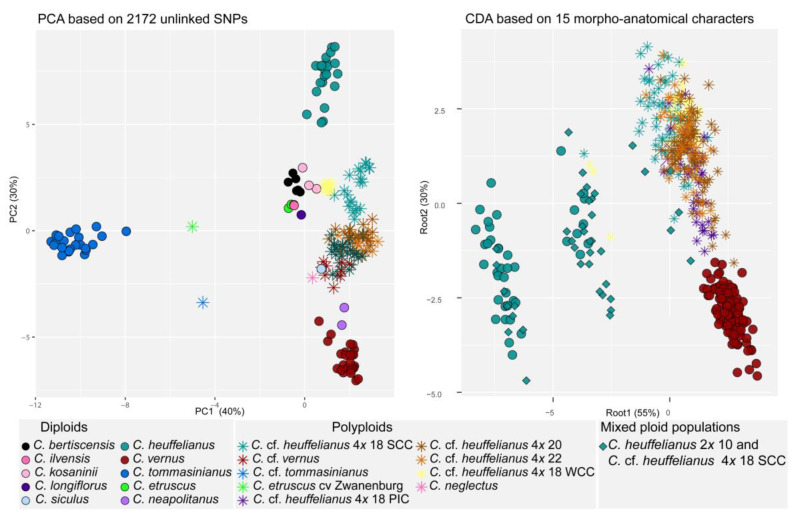
PCA based on 2127 unlinked GBS-derived SNPs (**left panel**) comprising all species of *Crocus* ser. *Verni*, and CDA based on 15 morpho-anatomical characters (**right panel**) of *C. vernus*, *C. heuffelianus,* and *C*. cf. *heuffelianus*.

**Figure 7 biology-12-00303-f007:**
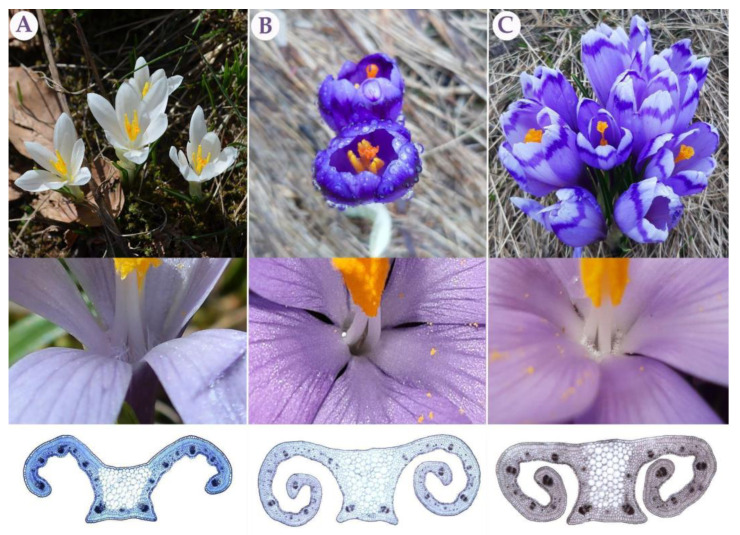
Flower, throat, and cross-section of leaf (from top to bottom) of parental species and a *C.* cf. *heuffelianus* polyploid: *C. vernus* (**A**), *C. heuffelianus* (**B**), and polyploid *C*. cf. *heuffelianus* representatives (**C**).

**Figure 8 biology-12-00303-f008:**
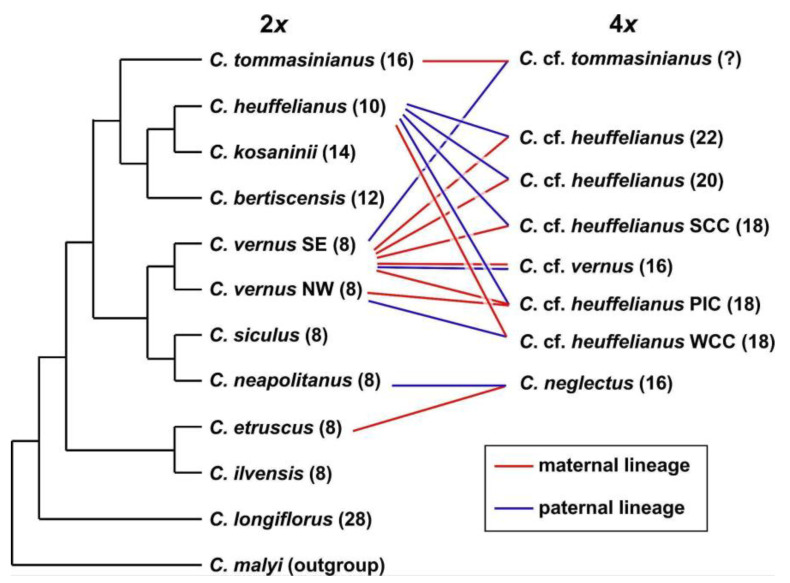
Schematic representation of species relationships in *Crocus* ser. *Verni* based on GBS, chloroplast, and nuclear single-copy gene data. For the tetraploids the maternal and paternal parents are indicated by lines connecting them to the respective diploid taxa. In brackets, chromosome numbers are provided.

**Table 1 biology-12-00303-t001:** Materials from *Crocus* ser. *Verni* used in this study (for an extended list see Appendix A).

Species	Individuals in Chloroplast/ GBS/NSCG ^1^/Morpho-Anatomical Analyses	Origin	*x*/Chromosome Number/Average 2C Genome Size
*Crocus bertiscensis* Raca, Harpke, Shuka, and V.Randjel.	4/6/1/0	Albania (ALB), Montenegro (MNE)	2*x*/12/6.66 pg
*Crocus etruscus* Parl.	2/3/0/0	Italy (ITA)	2*x*/8/7.58 pg
*Crocus ilvensis* Peruzzi and Carta	4/5/1/0	Italy (ITA)	2*x*/8/7.88 pg
*Crocus kosaninii* Pulević	2/3/2/0	Kosovo (XKX), Serbia (SRB)	2*x*/14/7.95 pg
*Crocus longiflorus* Raf.	2/1/2/0	Italy (ITA)	2*x*/28/3.21 pg
*Crocus neglectus* Peruzzi and Carta	1/1/1/0	Germany (GER)	4*x*/16/12.24 pg
*Crocus neapolitanus* (Ker Gawl.) Loisel.	3/2/1/0	Italy (ITA)	2*x*/8/–
*Crocus siculus* Tineo	0/1/0/0	Italy (ITA)	2*x*/8/–
*Crocus tommasinianus* Herb.	5/25/2/0	Bosnia and Herzegovina (BIH), Italy (ITA), Montenegro (MNE), Serbia (SRB)	2*x*/16/5.53 pg
*Crocus vernus* (L.) Hill	14/27/4/100	Albania (ALB), Bosnia and Herzegovina (BIH), France (FRA), Montenegro (MNE), Slovenia (SLO), Switzerland (CHE)	2*x*/8/5.78 pg
*Crocus heuffelianus* Herb.	8/22/4/70	Romania (ROU), Slovakia (SVK), Ukraine (UKR)	2*x*/10/7.73 pg
*Crocus* cf. *heuffelianus* (SCC)	7/19/1/70	Romania (ROU)	4*x*/18/12.84 pg
*Crocus* cf. *heuffelianus* (PIC)	5/26/1/40	Bosnia and Herzegovina (BIH), Slovenia (SLO)	4*x*/18/10.88 pg
*Crocus* cf. *heuffelianus* (WCC)	3/9/2/35	Slovakia (SVK)	4*x*/18/12.75 pg
*Crocus* cf. *heuffelianus*	12/23/0/80	Montenegro (MNE), Serbia (SRB)	4*x*/20/11.82 pg
*Crocus* cf. *heuffelianus*	3/14/1/40	Albania (ALB), Kosovo (XKX)	4*x*/22/11.95 pg
*Crocus* cf. *vernus*	5/9/1/0	Albania (ALB)	4*x*/16/12.38 pg
*Crocus* cf. *tommasinianus*	0/1/0/0	Montenegro (MNE)	4*x/-/-*
*Crocus malyi* Vis. (outgroup)	2/2/2/0	Croatia (HRV)	2*x*/30/–

^1^ NSCG = nuclear single-copy genes plus rDNA ITS.

**Table 2 biology-12-00303-t002:** Characteristics of the analyzed molecular datasets for *Crocus* ser. *Verni*.

	Chloroplast (*mat*K–*trn*Q + *ycf*1)	GBS Data 2*x*/2*x* + 4*x*	Nuclear Gene Regions *orcp*/*rcf*2/*topo*6/ITS
Number of sequences	81	93/194	37/42/39/22
Alignment lengths	4123	187,846	1160/568/819/646
Constant characters	3989	178,748/176,958	982/490/737/599
Variable characters	134	9098/10,888	178/78/82/47
Parsimony-informative characters	114	5899/6954	125/47/38/33
Number of MP trees	4	45/5400	142/10k/10k/10k ^1^
MP tree length	156	16,298/25,300	333/99/89/56
Consistency index (CI)	0.87	0.58/0.45	0.57/0.83/0.96/0.88
Retention index (RI)	0.98	0.83/0.77	0.66/0.88/0.97/0.93
Model of sequence evolution (BIC)	GTR + I	GTR + I + Γ	HKY + I + Γ/HKY + I + Γ/TrN + I/HKY + I

^1^ 10,000 trees were set as maxtree in MP analyses.

## Data Availability

The analyzed DNA sequences and raw reads are available through ENA study accession number PRJEB57934, Appendix A can be downloaded from e!DAL through https://doi.org/10.5447/ipk/2023/5, accessed on 10 February 2022.

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
