# Peer review of "Disentangling Crocus Series Verni and Its Polyploids"

_biology, 2023, doi:10.3390/biology12020303_

Round 1

Reviewer 1 Report

Raca et al. aimed at disentangling a polyploid complex within Crocus series Verni, using phylogenetic, phylogenomic, and morphological methods. Crocus is an economically (cultivars), but also an evolutionarily important and interesting genus (hybridization, polyploidy, disploidy). In detail, the authors conducted a multi-approach workflow based on chromosome counts, genome size measurements, genetic/genomic markers (chloroplast markers, nuclear single-copy genes, and GBS), and morphological/anatomical traits data. The authors are experts in these fields, and published many studies within the last ca. 10 years. Despite they gathered a lot of data and performed several analyses, many issues appeared while reading the manuscript. In general, a bit of background knowledge is missing in the introduction (and partly in Materials & Methods) to understand the theoretical framework and the biology of the Crocus species investigated in this study, which is crucial to follow the reconstruction of reticulate evolutionary relationships in this interesting genus. Many analyses have to be clarified in the context of polyploidy, hybridization, and disploidy, and partly (and potentially) some analysis re-done or new analysis performed. Before a more detailed review can take place (particularly the discussion part), a careful revision of the manuscript has to take place. I would suggest a major revision.

major and minor issues:

Title

The title is concise and provides the key result of the paper. However, it would be nice (for the readers) to include which polyploid complex is resolved within the paper.

Simple Summary

P. 1. L. 13: Please be more accurate concerning the definition of polyploidy: “..that are plants with more than two sets of chromosomes”.  Diploids also have multiple chromosomes, ie. 2.

P. 1. L. 17-18: ‘Among Crocus species, chromosome numbers…’ Please connect this sentence to the previous and subsequent one (I know your intention, but currently it is not clear why this information is necessary for this study; e.g., “In addition, evolutionary studies are even more complicated by disploidy in this group, i.e. ….).

P. 1. L. 19: I would delete the term “exemplarily” because it gives no information to the sentence.

P. 1. L. 20: Did you mean by “maternal- and biparental-inherited” apomictic versus sexual reproduction?

P. 1. L. 21: Please replace “polyploids” with “polyploid formation”. “Involved in polyploids” sounds a bit strange. And what does the “total evidence” approach mean? (see also abstract)

P. 1. L. 21: “Chromosome numbers partly changed..”

P. 1. L. 23: What are “diverse data”? Did you aim to say “multi-approach data”?

In general, it would be nice stating whether polyploids evolved via polyploidization (within species) or polyploidization and hybridization (allopolyploidy). I think that this should be one of the main messages of the study.

Abstract

P.1, L. 27-28: Is the group with unclear taxonomic relationships a species complex, or a species of different cytotypes? Please give more details (including the species/group name) here.

P.1, L. 29: I would suggest replacing “providing” with “leading to”. Does this process lead to intricate evolutionary patterns, or is this process in any way responsible for observed taxonomic problems?

P.1, L. 31-33: It’s an integrative taxonomic approach, right? So I should state it here. BTW: Is genomics used as the most important evidence in your study? If so, I would first say “phylogenetic analysis” and at the end “morphometry”. In addition, this sentence is very wordy and hard to read. Thus, I would suggest replacing “and geno-typing-by-sequencing-derived genome-wide distributed DNA polymorphisms” with “genome-wide GBS data (genotyping-by-sequencing)…”.

In general, you should add this information to the abstract: How many species did you incorporate into analyses? How many loci/chloroplast regions? Did species reproduce sexually, or apomictically (especially the polyploids?)?

P.1, L. 35-37: Significant correlation? (If yes, please give the p-value). Goes the correlation in a positive, or negative direction? Currently, it is not clear to me how the correlation between heterozygosity and genome size can distinguish di- and polyploids. Diploids usually have lower heterozygosity than polyploids, due to a combination of different parental species, or more alleles per locus that can develop in different ways (e.g., https://onlinelibrary.wiley.com/doi/full/10.1111/mec.15919). This comparison should be sufficient to distinguish di- and polyploids, or? In addition, how can a diploid species show a chromosome number range between 8 and 28? It might be the case that you explained it later on, but 90% of readers will first access the abstract and wonder about it.

Let’s assume the basic chromosome number of a diploid might be 8 (n=4), then 2n = 16 should already indicate tetraploids, (2n = 12 should be triploid, etc). You should add an explanation or at least an evolutionary explanation in brackets to clarify that.

Please state the basic chromosome number of the diploid level. Otherwise, the readers cannot follow.

P.1, L. 44-46: People always state that the described workflow of the study is kind of useful. Please be more specific about which kind of analysis/species group it is useful, and how it makes methodological progress.

Introduction

P. 2, L. 56: “while allopolyploids usually undergo…”

P. 2, L. 57-61: Please split the sentence for better comprehensibility (this is a nice and important explanation, but hard to read). In addition (and in general), please avoid the use of “could/would/should” in the MS as much as possible (it weakens your arguments/explanations). You might use instead probably/frequently/potentially/etc.

P. 2, L. 63-64: I think that you would like to point at “polyploidy is beneficial due to increased environmental flexibility”? (e.g., 10.1093/plcell/koaa015)

P. 2, L. 64-67: References are missing for statements.

P. 2, L. 70: “can be a source of information”.

To that point, I am missing a statement about the current lack of knowledge (far beyond the own model system where always information is missing) which motivated this study and which should be clearly pointed out in the introduction. For example, evolutionary, technical, workflow issues, etc. that have to be solved.

P. 2, L. 78-79: What do you mean by “…earlier approaches with molecular..”? Traditional molecular markers, such as ITS? Or population genetic markers such as microsatellites? Please be more specific and add the marker types in brackets.

P. 2, L. 80-90: Please give more details here about basic chromosome numbers, why diploids have chromosome numbers of a tetraploid (see abstract), and how this could be explained in an evolutionary context. Moreover, you state that C. heuffelianus has 2n = 10 -23, but are those all diploids (which I honestly do not believe) or are different ploidy levels involved? Particularly in this sentence, you refer to C. heufelianus “complex”, right? In the subsequent sentence, you are talking about the potential parental species, or? Or is C. neglectus simply a species within series Verni/heufelianus complex? Or in general, how does C. neglectus relate to this whole story?

In general, please state exactly which taxa the polyploids are, and which taxa the diploid parental progenitor species represent. Currently, I have no idea about which species’ evolutionary history you would like to reconstruct, and if it’s a diploid or polyploid (according to the introduction, it should be).

P. 3, L. 90-: I think that you intended to say “highly informative GBS data to reconstruct a robust phylogeny”, or? And in (ii), the focus is on inferring allo- vs. autopolyploidy, or?

Until now, you did not talk about taxonomic concepts, which is crucial for phylogenetic/taxonomic studies. Please give a few sentences within the introduction, which species concept you prefer (e.g., the latest “genetic lineage concept” based on de Queiroz ideas, or a genetic cluster concept where morphological data is only added as support? See for example 10.1002/tax.12365).

Material and Methods:

P.3., L.100-111: Please give a few details about sampling dates, locations, field vs garden, and what “/” should indicate (e.g., 24 populations C. heuffelianus/C. × heuffelianus à should that indicate mixed populations of C. heuffelianus and C. × heuffelianus?, and what is C. × heuffelianus? A hybrid of which species? à the same applies to C. vernus/C. × vernus). Please give more details about study taxa in the introduction (geography, morphotype, ploidy, etc) so that the reader can better follow/understand results about hybridization/polyploidization events among species.  This lack of knowledge makes understanding reticulate evolutionary processes in the C. heuflianus complex almost impossible.

Moreover, please add GPS coordinates to all tables including location information (e.g., Table 1, S1, S2)

P.3., L.121: Citation of Geneious is missing. Where are genetic/genomic raw data deposited (deposition is usually mandatory)?

P.3., L.125-126: “with one of the latter included twice as a replicate”, why?

In general, please add version numbers to all programs and citations! (missing many times)

P.3., L.135-148: You cannot simply take IPYRAD default settings for creating genomic alignments. Settings have a tremendous influence on under- and overmerging of reads into loci (and loci into final alignments; orthologs vs. paralogs) and thus final alignments (number of loci, number of SNPs, missing data percentage, etc., see 10.1002/tax.12365, 10.1111/mec.15919). Please give details about your final ipyrad output filter settings (number of loci, SNPs, missing data), and make sure that different clustering thresholds (you only used 0.85 default) have no tremendous influence on final alignment characteristics and thus phylogenetic results.

In addition, which ploidy setting did you choose? For instance, specifying ploidy = 2 would mean that all alleles above two are filtered out (bad for polyploids). It is also recommended by IPYRAD to do separate parameter optimization for di- and polyploids, and merge both assemblies in IPYRAD step 6. That is due to the fact that different ploidy levels have different genetic similarities within and across loci, and thus affect clustering threshold, assemblies, and final results (see again 10.1002/tax.12365, 10.1111/mec.15919). Please make sure that you handled different ploidy types correctly within analyses.

Moreover, why did you use fastSTRUCTURE? It has several known bugs (I also experienced a lot) and is not adapted to RAD-Seq/GBS data (https://github.com/rajanil/fastStructure/issues). You already cited LEA – it has a nice, fast, and STRUCTURE-like approach called “sNMF” which is robust to moderate numbers of missing data and different ploidy levels (which is beneficial for your study).

P. 4-5: Please present chromosome counts and genome size first, because you have to give information about ploidy in subsequent IPYRAD analyses.

P. 4, L. 149-152: FST values, as far as I know, can only be calculated for diploids due to Hardy-Weinberg assumptions. How did you do that here? Please give more details. Or did you only specify ploidy = 2 in IPYRAD analysis? However, this would lead to erroneous heterozygosity estimations of polyploids, and results…

P. 5, L. 173: And which coverage you are aiming at, for diploids and polyploids? The number of reads alone bears not much information.

P. 5, L. 177-178, and in general: Why did you not use GBS data for reconstructing the evolutionary history of allopolyploids as well? RADPainter+fineRADstructure, or PhyloNetworks can handle GBS data, and are able to inform about reticulate evolution.

Please give estimated ploidy levels in Table S2.

P. 5, L. 191: And where from comes the information of C. Siculus?

P. 5, L. 198: Internal, or external standard?

P. 5, L. 199: “Genome size measurements aimed at identifying...”

P. 5, L. 200: Please rephrase the entire sentence for better comprehensibility. Did you mean that you did not measure the same individuals within FC and molecular analyses? That’s bad…particularly in a group of varying chromosome numbers and disploidy. In how many cases did the samples overlap? Please provide information on which samples are used in which analyses so that the reader can easily access it.

P. 6-7: Which morphological traits were evaluated? Are they taxonomically informative? (please cite literature, or add a paragraph to the introduction about which traits are used for taxonomic classification within this group). Moreover, are all traits continuous (if not, how did you treat ordinal/categorical data in PCA/DCA?)? Did you standardize all traits before running PCA? Is a PCA suitable here (e.g., gradient length < 2.5?)? Maybe an evolutionary PCA is more appropriate (https://search.r-project.org/CRAN/refmans/adiv/html/evopca.html ).

Results:

P. 7, L. 275-289: Again (see abstract), please provide p values and statistics (and partly figures) for these comparisons/relationships (which correlation test did you use? à has to be specified in the M&M part). You described comparisons/relationships very well, but statistics are needed to confirm differences. In addition, the theory about ploidy and genome-wide heterozygosity is missing in the introduction, important to interpret and discuss the results later on.

P. 8-12, Figure legends: Fig. 1-5 Please specify the used alignment, marker type, and number of loci in the figure legend. Fig. 1 Again, how can 2x = 28 be diploid? If x = 4 in Crocus, 2x = 28 should be ca. hexaploid. More information has to be given in the legend. Fig. 3, and partly Fig. 5: This is a very important issue. According to the intro, we assume allopolyploid relationships, and thus reticulate evolution. Reticulate evolution hurts the assumptions of a phylogenetic tree (i.e., bifurcating patterns). Thus, you need to find another analysis to evaluate all ploidy levels together (e.g., RADpainter, sNMF, etc.). I would thus recommend shifting this tree to the Supplements, or deleting it.

Fig. 2-3. Support values are missing.

Fig. 5. Please give (a) and (b) here. Does the figure only illustrate two different single-copy genes? How did you combine information for allopolyploid reconstruction in case gene 1 shows a different pattern than gene 2? Either simplify the figure, or put it in the supplements. Currently, this looks very confusing to me.

Where is the allopolyploid reconstruction (text, and figures)? It seems to be missing in the results section…

P.12-13, L. 389-: In this study, you are doing phylogenetic/-omic analysis based on several taxa, which is clearly no population genetics. Therefore, it is very strange that you label the subsequent paragraph with “population genetic analysis”. You could name it “Phylogenomic data analysis”.

Why did you show a PCA based on GBS? It isn’t specified within the M&M, and also a bit strange because it only shows clustering and not genomic composition, and a bunch of better evaluation methods is available for phylogenomics (e.g., RADPainter, sNMF, PhyloNetworks,…).

Fig. 6. Explained axis percentages are missing. (b) arrows are missing that indicate in which direction a morphological trait is increasing/decreasing, and factor strength (i.e., length of arrow).

P.14, L. 440-454: How many clusters are found? Do they correspond to genetic clusters? Which are the most important traits? Which cluster is characterized by which trait combinations? Moreover, this is only descriptive. Please run statistical tests (e.g., ANOVA/Kruskal-Wallis-Tests + posthoc tests) to show significant differences among clusters/taxa.

Figure 7 should be shifted to the M&M, because it contains no results and should rather inform morphometric data analysis, or? Otherwise, please highlight morphological differences in this figure with arrows, etc.

Author Response

Title

The title is concise and provides the key result of the paper. However, it would be nice (for the readers) to include which polyploid complex is resolved within the paper.

Title was slightly changed.

Simple Summary

  1. 1. L. 13: Please be more accurate concerning the definition of polyploidy: “..that are plants with more than two sets of chromosomes”. Diploids also have multiple chromosomes, ie. 2.

Multiple chromosomes, of course. But not multiple sets of chromosomes. So the term seems right as written.

  1. 1. L. 17-18: ‘Among Crocus species, chromosome numbers…’ Please connect this sentence to the previous and subsequent one (I know your intention, but currently it is not clear why this information is necessary for this study; e.g., “In addition, evolutionary studies are even more complicated by disploidy in this group, i.e. ….).

Done

  1. 1. L. 19: I would delete the term “exemplarily” because it gives no information to the sentence.

Done

  1. 1. L. 20: Did you mean by “maternal- and biparental-inherited” apomictic versus sexual reproduction?

Why apomictic? We never mention apomixis even once throughout the entire manuscript. Chloroplasts are in most angiospems provided by the mother (i.e. maternal) while nuclear DNA is derived from both parents (i.e. biparental).

Expanded.

  1. 1. L. 21: Please replace “polyploids” with “polyploid formation”. “Involved in polyploids” sounds a bit strange. And what does the “total evidence” approach mean? (see also abstract)

Done

  1. 1. L. 21: “Chromosome numbers partly changed..”

Done

  1. 1. L. 23: What are “diverse data”? Did you aim to say “multi-approach data”?

Changed

In general, it would be nice stating whether polyploids evolved via polyploidization (within species) or polyploidization and hybridization (allopolyploidy). I think that this should be one of the main messages of the study.

Wording changed

Abstract

P.1, L. 27-28: Is the group with unclear taxonomic relationships a species complex, or a species of different cytotypes? Please give more details (including the species/group name) here.

Yeah, that’s the question here. And our analyses were made to be able to clarify it in the end. Depending on authors, nearly the entire series was treated either as a single species with different subspecies, an aggregate, different species, different diploid species with tetraploid cytotypes, …  We assume they are species at different ploidy levels. However, taxonomic conclusions will only be made when we also understand reproductive isolation between the ‘lineages/cytotypes.’ Thus, currently it would be premature to use more exact terms.

  1. heuffelianus, where most of the polyploids were thought to belong, is now mentioned.

P.1, L. 29: I would suggest replacing “providing” with “leading to”. Does this process lead to intricate evolutionary patterns, or is this process in any way responsible for observed taxonomic problems?

Changed

P.1, L. 31-33: It’s an integrative taxonomic approach, right? So I should state it here. BTW: Is genomics used as the most important evidence in your study? If so, I would first say “phylogenetic analysis” and at the end “morphometry”. In addition, this sentence is very wordy and hard to read. Thus, I would suggest replacing “and geno-typing-by-sequencing-derived genome-wide distributed DNA polymorphisms” with “genome-wide GBS data (genotyping-by-sequencing)…”.

Changed

In general, you should add this information to the abstract: How many species did you incorporate into analyses? How many loci/chloroplast regions? Did species reproduce sexually, or apomictically (especially the polyploids?)?

Changed

P.1, L. 35-37: Significant correlation? (If yes, please give the p-value). Goes the correlation in a positive, or negative direction? Currently, it is not clear to me how the correlation between heterozygosity and genome size can distinguish di- and polyploids. Diploids usually have lower heterozygosity than polyploids, due to a combination of different parental species, or more alleles per locus that can develop in different ways (e.g., https://onlinelibrary.wiley.com/doi/full/10.1111/mec.15919).

For these crocuses: Yes.  Generally for all plants: No (here breeding system, linkage disequilibrium, and differences in population’s genetic diversity together with gene flow within and among the population(s) plays a role). We didn’t use “significant”; see comment further below.

This comparison should be sufficient to distinguish di- and polyploids, or?

No, not when the correlation isn’t established first through comparisons between genome size and observed heterozygosity.

In addition, how can a diploid species show a chromosome number range between 8 and 28? It might be the case that you explained it later on, but 90% of readers will first access the abstract and wonder about it.

Good, might get them interested (that’s the same reaction we had). Here we can only state that it is like this. We are currently working on why and how.

Let’s assume the basic chromosome number of a diploid might be 8 (n=4), then 2n = 16 should already indicate tetraploids, (2n = 12 should be triploid, etc). You should add an explanation or at least an evolutionary explanation in brackets to clarify that.

Such an explanation would be very speculative at the moment. The different possibilities are mentioned in the discussion (dysploidy and/or polyploidization with massive genome downsizing).

Please state the basic chromosome number of the diploid level. Otherwise, the readers cannot follow.

In contrast to many other plant groups this is not possible for this and other Crocus series as dysploidy is masking n.

P.1, L. 44-46: People always state that the described workflow of the study is kind of useful. Please be more specific about which kind of analysis/species group it is useful, and how it makes methodological progress.

All. Combined.  Single ones don’t help.

Introduction

  1. 2, L. 56: “while allopolyploids usually undergo…”

Changed

  1. 2, L. 57-61: Please split the sentence for better comprehensibility (this is a nice and important explanation, but hard to read). In addition (and in general), please avoid the use of “could/would/should” in the MS as much as possible (it weakens your arguments/explanations). You might use instead probably/frequently/potentially/etc.

Changed

  1. 2, L. 63-64: I think that you would like to point at “polyploidy is beneficial due to increased environmental flexibility”? (e.g., 10.1093/plcell/koaa015)

We now refer to this newer review instead of Yves’ earlier original paper

  1. 2, L. 64-67: References are missing for statements.

Added

  1. 2, L. 70: “can be a source of information”.

To that point, I am missing a statement about the current lack of knowledge (far beyond the own model system where always information is missing) which motivated this study and which should be clearly pointed out in the introduction. For example, evolutionary, technical, workflow issues, etc. that have to be solved.

Yes, but we provide here the Intro to the mechanism behind the problem we want to tackle. In the following paragraph is the Intro to the organisms and what information is lacking, before we then describe the intended study. Seems logical to us, particularly as there are a lot of things not understood in polyploidization, which we don’t want to answer with this study.

  1. 2, L. 78-79: What do you mean by “…earlier approaches with molecular..”? Traditional molecular markers, such as ITS? Or population genetic markers such as microsatellites? Please be more specific and add the marker types in brackets.

But this is about phylogenies (“badly resolved phylogenies”) not population-level studies within species. Thus, it should be clear from the context.

  1. 2, L. 80-90: Please give more details here about basic chromosome numbers, why diploids have chromosome numbers of a tetraploid (see abstract), and how this could be explained in an evolutionary context.

See above.

Moreover, you state that C. heuffelianus has 2n = 10 -23, but are those all diploids (which I honestly do not believe) or are different ploidy levels involved?

This information is provided in the next sentence where we refer to an earlier study and what’s known or not.

Particularly in this sentence, you refer to C. heufelianus “complex”, right?

Up to now C. heuffelianus is mostly treated as a species. But of course it is a polyploid complex, something that we clarify in the context of the described study/manuscript.

In the subsequent sentence, you are talking about the potential parental species, or? Or is C. neglectus simply a species within series Verni/heufelianus complex? Or in general, how does C. neglectus relate to this whole story?

As written, this is the other species in the series that was already defined as polyploid.

Sentences now re-ordered to possibly avoid confusion.

In general, please state exactly which taxa the polyploids are, and which taxa the diploid parental progenitor species represent.

This is at the center of the analyses we conducted and describe here. We can’t tell all the secrets already at the beginning.

Currently, I have no idea about which species’ evolutionary history you would like to reconstruct, and if it’s a diploid or polyploid (according to the introduction, it should be).

If we would detail all the species already here you would probably accuse us of being repetitive because this is described in the next paragraph (Materials). We have to provide the framework (what group, what’s known) of the study in Intro, not all the details which belong to Materials and Methods.

  1. 3, L. 90-: I think that you intended to say “highly informative GBS data to reconstruct a robust phylogeny”, or? And in (ii), the focus is on inferring allo- vs. autopolyploidy, or?

Changed.

Until now, you did not talk about taxonomic concepts, which is crucial for phylogenetic/taxonomic studies. Please give a few sentences within the introduction, which species concept you prefer (e.g., the latest “genetic lineage concept” based on de Queiroz ideas, or a genetic cluster concept where morphological data is only added as support? See for example 10.1002/tax.12365).

No. We don’t want to tread into the muddy waters of species concepts. At least not here. Though, this might be a topic in a taxonomic revision of the polyploid taxa of the heuffelianus complex. Here, as written in the manuscript (“intend to understand the evolution of Crocus series Verni with particular reference to the origin of the polyploid species and cytotypes”), we want to provide the basis for further work in series Verni.

Material and Methods:

P.3., L.100-111: Please give a few details about sampling dates, locations, field vs garden, and what “/” should indicate (e.g., 24 populations C. heuffelianus/C. × heuffelianus à should that indicate mixed populations of C. heuffelianus and C. × heuffelianus?, and what is C. × heuffelianus? A hybrid of which species? à the same applies to C. vernus/C. × vernus). Please give more details about study taxa in the introduction (geography, morphotype, ploidy, etc) so that the reader can better follow/understand results about hybridization/polyploidization events among species.  This lack of knowledge makes understanding reticulate evolutionary processes in the C. heuflianus complex almost impossible.

One can easily overload the reader with hard-to-read information that is better presented in a table. These demands look very much to result in information overload. We ‘ve already changed “x” to “cf.” as more suitable according to Editor’s suggestions and explained the new terminology in Intro.

Regarding “/”: Currently in both species the polyploids are still taxonomically subsumed belonging to the diploid taxon. Therefore, we refer to them together. Details are listed in the tables.

Moreover, please add GPS coordinates to all tables including location information (e.g., Table 1, S1, S2)

We are generally very reluctant to provide exact origin data, particularly GPS coordinates. Crocuses are nice ornamentals and poaching, resulting in population extinctions, happened several times when we published this information. Therefore we give country and locality but no coordinates. If somebody wants to harvest the plants he/she has to at least travel there and hike around and search the area.

P.3., L.121: Citation of Geneious is missing. Where are genetic/genomic raw data deposited (deposition is usually mandatory)?

Geneious is distributed through the company Geneious. Thus, it's kind of self-explanatory. Data availability is stated in the “Data Availability Statement”.

P.3., L.125-126: “with one of the latter included twice as a replicate”, why?

Usual procedure to see how reproducible the lab protocols resulting in data/sequences are.

In general, please add version numbers to all programs and citations! (missing many times)

Given the first time the program is mentioned, afterwards we refer to the program.

P.3., L.135-148: You cannot simply take IPYRAD default settings for creating genomic alignments. Settings have a tremendous influence on under- and overmerging of reads into loci (and loci into final alignments; orthologs vs. paralogs) and thus final alignments (number of loci, number of SNPs, missing data percentage, etc., see 10.1002/tax.12365, 10.1111/mec.15919). Please give details about your final ipyrad output filter settings (number of loci, SNPs, missing data), and make sure that different clustering thresholds (you only used 0.85 default) have no tremendous influence on final alignment characteristics and thus phylogenetic results.

We totally agree here! During the course of the study several settings (thresholds, proportion of shared polymorphic sites in a locus etc.) were tested and we also used other tools/approaches (STACKS, an own pipeline similar to ddocent). Output files like the vcf files were checked, reads were also mapped to loci, which had a relatively high number of SNPs, and checked to see if they are potential paralogs (however, diploids were found to be homozygous or if heterozygous then with max. 2 alleles).Finally, the default clustering threshold performed well to recover a sufficient number of loci across the whole series, same for the maximum number of shared heterozygous sites to filter out paralogs. At the end we arrived at the default settings working best (good work by Eaton and co!).

Number of loci, missing data etc. is given, e.g., in 3.3. or partly also in the respectives figures.

In addition, which ploidy setting did you choose? For instance, specifying ploidy = 2 would mean that all alleles above two are filtered out (bad for polyploids). It is also recommended by IPYRAD to do separate parameter optimization for di- and polyploids, and merge both assemblies in IPYRAD step 6. That is due to the fact that different ploidy levels have different genetic similarities within and across loci, and thus affect clustering threshold, assemblies, and final results (see again 10.1002/tax.12365, 10.1111/mec.15919). Please make sure that you handled different ploidy types correctly within analyses.

Most of the SNPs are bi-allelic in the polyploids in our data set (was checked in the vcfs files after running ipyrad with different settings regarding the maximum number of alleles plus by comparing the number of loci as well as by mapping).  Also the ddocent similar approach (first generating a pseudo-reference using only diploids, then mapping all samples to that reference) was free assumptions on minimum number of alleles per SNP per loci. Furthermore, only around 2.5% of loci of e.g. C. vernus and C. heuffelianus constituted more than 2 possible alleles over all samples of these species. In our finally used data set, using 2 or 4 for ploidy resulted in the identical number of loci.

Moreover, why did you use fastSTRUCTURE? It has several known bugs (I also experienced a lot) and is not adapted to RAD-Seq/GBS data (https://github.com/rajanil/fastStructure/issues). You already cited LEA – it has a nice, fast, and STRUCTURE-like approach called “sNMF” which is robust to moderate numbers of missing data and different ploidy levels (which is beneficial for your study).

We normally use at least two assignment programs as all of them have certain issues.

  1. 4-5: Please present chromosome counts and genome size first, because you have to give information about ploidy in subsequent IPYRAD analyses.

But then we would have to do this already before we mention materials (2.1) because there we refer to some 2n = 18 karyotypes without explaining how we inferred this. Therefore we hope readers are able to prescind a bit. In Results we follow the more logical order (chromosomes - ploidy - phylogeny - populations)

  1. 4, L. 149-152: FST values, as far as I know, can only be calculated for diploids due to Hardy-Weinberg assumptions. How did you do that here? Please give more details. Or did you only specify ploidy = 2 in IPYRAD analysis? However, this would lead to erroneous heterozygosity estimations of polyploids, and results…

It is possible, because most SNPs are bi-allelic in the polyploids.

  1. 5, L. 173: And which coverage you are aiming at, for diploids and polyploids? The number of reads alone bears not much information.

No, number of height depths cluster i.a. loci produced by the cutting enzymes is needed to calculate the app./average coverage (added). We target a coverage of 30-100x.

  1. 5, L. 177-178, and in general: Why did you not use GBS data for reconstructing the evolutionary history of allopolyploids as well? RADPainter+fineRADstructure, or PhyloNetworks can handle GBS data, and are able to inform about reticulate evolution.

Not really well. For example, RADPainter+fineRADstructure didn’t identify the allopolyploids at all (similar to the ancestral population assignment tools used).

These programs claim they can do it, but if you use diverse analysis tools they are not congruent.

Please give estimated ploidy levels in Table S2.

Comment is unclear, Supplementary Table S2 includes PCR conditions and primer information.

  1. 5, L. 191: And where from comes the information of C. Siculus?

We don’t give genome sizes for C. siculus and C. neapolitanus (Table 1), as we don’t have them.

  1. 5, L. 198: Internal, or external standard?

Don’t get this question. It’s always an internal standard that is co-chopped with the target individuum. External standards might only really work for ploidy determination not for genome size measurements.

  1. 5, L. 199: “Genome size measurements aimed at identifying...”

Changed.

  1. 5, L. 200: Please rephrase the entire sentence for better comprehensibility. Did you mean that you did not measure the same individuals within FC and molecular analyses? That’s bad…particularly in a group of varying chromosome numbers and disploidy. In how many cases did the samples overlap? Please provide information on which samples are used in which analyses so that the reader can easily access it.

But we write that we use wherever possible the very same individual for DNA extraction and FC. In rare cases when this was not possible we used several plants from the same population. With this we control for uniformity in the population. Moreover, in all cases we would immediately detect outliers in the GBS analyses, where individuals from polyploid populations group together separate from the diploid individuals and are characterized by their Ho class. In cases where we had mixed di- and tetraploid populations only FC-measured individuals made it into GBS.

  1. 6-7: Which morphological traits were evaluated? Are they taxonomically informative? (please cite literature, or add a paragraph to the introduction about which traits are used for taxonomic classification within this group). Moreover, are all traits continuous (if not, how did you treat ordinal/categorical data in PCA/DCA?)? Did you standardize all traits before running PCA? Is a PCA suitable here (e.g., gradient length < 2.5?)? Maybe an evolutionary PCA is more appropriate (https://search.r-project.org/CRAN/refmans/adiv/html/evopca.html ).

We add citations of the relevant literature for this group to M&M part. The total list of characters related with the morphology and leaf anatomy is presented in the Supplementary Table S4 (the abbreviation legend is given at the end of the Table). The qualitative characters were converted to numbers of course. The main reason we used the PCA was just to highlight really significant features we could furthermore use in CDA.

Results:

  1. 7, L. 275-289: Again (see abstract), please provide p values and statistics (and partly figures) for these comparisons/relationships (which correlation test did you use? à has to be specified in the M&M part). You described comparisons/relationships very well, but statistics are needed to confirm differences. In addition, the theory about ploidy and genome-wide heterozygosity is missing in the introduction, important to interpret and discuss the results later on.

What test do you need to see that all FC genome-size measured diploids have a low Ho, while the FC genome-size measured tetraploids have a high Ho and in between these data clusters is a gap where none of them falls in. One is allowed to speak about correlations without this test. I would always ask for a test if somebody writes about significant relationships/correlations.

  1. 8-12, Figure legends: Fig. 1-5 Please specify the used alignment, marker type, and number of loci in the figure legend.

Done

Fig. 1 Again, how can 2x = 28 be diploid? If x = 4 in Crocus, 2x = 28 should be ca. hexaploid.

No, it is not. Also the outgroup (C. malyi) with 30 chromosomes is a nice diploid with bivalents. Thus, … But this was already answered in the beginning (dysploidy)

More information has to be given in the legend. Fig. 3, and partly Fig. 5:

Done

This is a very important issue. According to the intro, we assume allopolyploid relationships, and thus reticulate evolution. Reticulate evolution hurts the assumptions of a phylogenetic tree (i.e., bifurcating patterns). Thus, you need to find another analysis to evaluate all ploidy levels together (e.g., RADpainter, sNMF, etc.). I would thus recommend shifting this tree to the Supplements, or deleting it.

We see in these trees if certain polyploids group together and apart from diploid progenitors or together with them. Particularly MP is rather sensitive to reticulations (hybrids and allopolyploids group normally at the base/as sister to one of  their diploid progenitors) and is a good indicator of gene flow. Therefore the methods we use to detect and discern the different groups of tetraploids is valid. Particularly as we don’t claim that these trees reflect the true phylogeny of the group. This we summarize in Fig. 8.

Fig. 2-3. Support values are missing.

These are the schemes illustrating the detailed trees given in the supplements. They provide information on the groups found and how chloroplast types are distributed in the di- and polyploid accessions. Thus, support values in the supplemental materials should be enough.

Fig. 5. Please give (a) and (b) here. Does the figure only illustrate two different single-copy genes? How did you combine information for allopolyploid reconstruction in case gene 1 shows a different pattern than gene 2? Either simplify the figure, or put it in the supplements. Currently, this looks very confusing to me.

We prefer to directly provide the marker names instead of (a) and (b). Yes, each tree is only showing one nuclear single-copy marker. We have some cases in our data set, where the position in the GBS tree and the cp haplotype alone are not sufficient to conclude on the parental species (some group as sister to their maternal parent also in the GBS tree). This two exemplary trees are crucial to see (i) to which diploid the alleles of these allotetraploids are grouping and (ii) both tree display that we don’t only have incongruences between cp and GBS data set, but also between single gene trees and that their high HI within the GBS data set is probably caused by loci reflecting different evolutionary histories (incongruent gene tree, possibly caused by ILS and hybridization). The figure is already simplified by collapsing clades.  (Seems with the tree cartoons before we spoiled the reviewer :-)

Where is the allopolyploid reconstruction (text, and figures)? It seems to be missing in the results section…

It is in the discussion (Fig. 8) since it is part of the conclusion of all our analyses.

P.12-13, L. 389-: In this study, you are doing phylogenetic/-omic analysis based on several taxa, which is clearly no population genetics. Therefore, it is very strange that you label the subsequent paragraph with “population genetic analysis”. You could name it “Phylogenomic data analysis”.

Changed.

Why did you show a PCA based on GBS? It isn’t specified within the M&M, and also a bit strange because it only shows clustering and not genomic composition, and a bunch of better evaluation methods is available for phylogenomics (e.g., RADPainter, sNMF, PhyloNetworks,…).

It compares directly to the morphometric data shown beside, showing the suitability of morphological characters that finally allow to even distinguish the diploid C. heuffelianus from the allotetraploid C. cf. heuffelianus. It is described in Material and Methods. In the PCA eigenvectors and later on eigenvalues are inferred based on genomic compositions. Here, it nicely placed the allotetraploids in between the parental species, while tools like RADPainter or sNMF/lNMF as implemented in LEA failed. As such the genomic composition is better reflected in the PCA. Clustering methods like those implemented in LEA, STRUCTURE etc. don’t necessarily reflect genomic composition due to their methodology.

Fig. 6. Explained axis percentages are missing. (b) arrows are missing that indicate in which direction a morphological trait is increasing/decreasing, and factor strength (i.e., length of arrow).

The percentages can be derived from the Supplementary Table S8 (see the last row “Prp. Totl.” - a cumulative proportion - means that first two axes are carrying 85% of the total variability: Root1 55%, Root2 30%. The direction of morphological features can also be seen in Supplementary Table S8 (sign “-” is indicating the negative correlation with specific axis), as well as the factor strength (with a threshold of 0.7).

P.14, L. 440-454: How many clusters are found? Do they correspond to genetic clusters? Which are the most important traits? Which cluster is characterized by which trait combinations? Moreover, this is only descriptive. Please run statistical tests (e.g., ANOVA/Kruskal-Wallis-Tests + posthoc tests) to show significant differences among clusters/taxa.

Grouping of allotetraploids and important traits (please see Chapter 3.4. of Results) are mentioned in the manuscript and discussed in the context with molecular results. Our own experience confirmed that such a tests are “not so sensitive” (e.g. every single character is significant :). So we used a PCA as the main statistical test this time (to highlight really significant characters). Finally, PCA would never highlight the characters that are listed as insignificant with any single statistical test; That ‘s why we think including even the common statistical tests would be redundant and we would really prefer to avoid it.

Figure 7 should be shifted to the M&M, because it contains no results and should rather inform morphometric data analysis, or? Otherwise, please highlight morphological differences in this figure with arrows, etc.

But it illustrates the main differences between the types that were identified with the statistical analyses of the morphology and explained in the text (in context with Fig. 6 and results of CDA).

Reviewer 2 Report

Disentangling a polyploid complex within Crocus series Verni

As a reviewer I have following suggestion/amendments:

-kindly use alternate world for Disentangling, as suitable (a suggestion)

Abstract

-write in one format e.g. (2C = 3.21 pg.), like, Tetraploid genomes have 2C sizes of 10.88 pg. to 12.84 pg.

- you used many approaches and give only Heterozygosity distribution correlated results, give DNA base phylogeny, how about chloroplast and other techniques.

Introduction

Try to write in one format, [16,18]. or Harpke et al. [15]

Here we intend to understand the evolution of Crocus,

-what is concerned of ploidy in evolution, ploidy is different from evolution please correct in last paragraph.

2. Materials and Methods

2.1. Plant materials

Our study includes plants from 63 populations: 24 C. heuffelianus/C. ×heuffelianus, 13 etc.

If these are 63 populations and then how many samples you draw from these, explain or write 63 samples from one or many species.

Table 1. make separate rows and columns, starting with serial number to easy understand.

DNA extraction and Sanger sequencing

Kindly give number of samples or detail of samples, later, we may understand the results

Give pics of gel, possible

Chromosome counts

Any image.

Give detail of samples.

Chromosome counts

Any picture of cytology/anatomical. How about stomata?? more precise.

Genome size measurements

Please use your diploid genomic DNA from your research, not from the other crops as reference/standard. 

3. Results

Write in same format (2n = 18, 20, 22).

GBS-derived heterozygosity to ploidy

Please write the names of some species/samples etc.

Also write the number of samples/individuals/specie along with results, falling in different categories.

Genome sizes

Chromosome counts showed a general negative relationship between genome size and chromosome number in both diploid and tetra-ploid taxa. This relationship is weaker when C. longiflorus is excluded (Supplementary Fig. 277 S1).

-This can be corrected by using, diploid (2X) samples from your population as reference/standard and then comparted your all samples.  

GBS-derived heterozygosity to ploidy

As a con-sequence, all samples with a H0 below 0.03 are considered to represent diploids, while all samples with a H0 above 0.035 are considered polyploids.

-Kindly write the names of samples or their numbers in your results.

3.2. Phylogenetic inference

GBS-derived data

You may also describe your data into groups and divers’ individuals in Phylogenetic tree, strict consensus tree and strict consensus tree, as you do in chloroplast regions.

Discussions

Figure 8. add this to your results as combine study, and explain a little or main points/same in discussion.

4.2. General results regarding phylogeny and systematics- 4.3. Chromosome and genome size evolution, remove this, make as one part as discussion, its my suggestion.

Overall, your results are excellent with good effort and approach.

Good Luck

Author Response

-kindly use alternate world for Disentangling, as suitable (a suggestion)

No. It describes what we are doing.

Abstract

-write in one format e.g. (2C = 3.21 pg.), like, Tetraploid genomes have 2C sizes of 10.88 pg. to 12.84 pg.

- you used many approaches and give only Heterozygosity distribution correlated results, give DNA base phylogeny, how about chloroplast and other techniques.

Done.

Introduction

Try to write in one format, [16,18]. or Harpke et al. [15]

These are 2 different kinds of references, a) more general ones and b) where we refer in the specific context to certain authors (like a paper where we used a certain protocol). Therefore we think this is valid (and usual, if you check other publications in Biology) and grammatically correct: Sounds awkward to us to write, e.g. “we used the protocol described by [15]”.

Here we intend to understand the evolution of Crocus,

-what is concerned of ploidy in evolution, ploidy is different from evolution please correct in last paragraph.

The main topic of our paper is that polyploidization and evolution are intermingled (see Introduction). Thus, the statement seems correct to us as is.

  1. Materials and Methods

Plant materials

Our study includes plants from 63 populations: 24 C. heuffelianus/C. ×heuffelianus, 13 etc.

If these are 63 populations and then how many samples you draw from these, explain or write 63 samples from one or many species.

Table 1. make separate rows and columns, starting with serial number to easy understand.

No, won’t make anything clearer and would not follow the journal’s format.

DNA extraction and Sanger sequencing

Kindly give number of samples or detail of samples, later, we may understand the results

Give pics of gel, possible

Number of samples was and still is given in Table 1 and in more detail in the Supplementary Table S1. Sequences were submitted to ENA and are openly available. Gel pictures for extracted genomic DNA? Really?   

Chromosome counts

Any image.

Give detail of samples.

Chromosome images were already presented in Supplementary Figure S1.

Chromosome counts

Any picture of cytology/anatomical. How about stomata?? more precise.

These are already exemplarily provided in Figure 7. Chromosome images are already presented in Supplementary Figure S1. We did not measure stomata sizes, as we have genome size data and stomata size classes are very variable with regard to ploidy level.

Genome size measurements

Please use your diploid genomic DNA from your research, not from the other crops as reference/standard.

This statement is contradicting whatever is best practice in genome-size estimating for the last two decades (see the plethora of work by Greilhuber). Cultivars have a very narrow gene pool and are uniform regarding their genome size and are therefore ideal as standards (i.e. comparable among labs), which isn’t the case for wild crocuses. Moreover, we measure with PI not DAPI so that differences in genome composition don’t play a role.

  1. Results

Write in same format (2n = 18, 20, 22).

Revised.

GBS-derived heterozygosity to ploidy

Please write the names of some species/samples etc.

Also write the number of samples/individuals/specie along with results, falling in different categories.

Done

Genome sizes

Chromosome counts showed a general negative relationship between genome size and chromosome number in both diploid and tetra-ploid taxa. This relationship is weaker when C. longiflorus is excluded (Supplementary Fig. 277 S1).

-This can be corrected by using, diploid (2X) samples from your population as reference/standard and then comparted your all samples. 

No, this statement of the reviewer is wrong (see above).

GBS-derived heterozygosity to ploidy

As a con-sequence, all samples with a H0 below 0.03 are considered to represent diploids, while all samples with a H0 above 0.035 are considered polyploids.

-Kindly write the names of samples or their numbers in your results.

We listed the species with Ho below 0.03 and above 0.035 in the manuscripts. Details can be found in the Supplementary Table S5, to which we refer in the text.

3.2. Phylogenetic inferences

GBS-derived data

You may also describe your data into groups and divers’ individuals in Phylogenetic tree, strict consensus tree and strict consensus tree, as you do in chloroplast regions.

This comment is unclear to us.

Discussions

Figure 8. add this to your results as combine study, and explain a little or main points/same in discussion.

But this illustrates our conclusion and is not somehow calculated by phylogenetic algorithms. Thus, the position within Discussion feels right to us.

4.2. General results regarding phylogeny and systematics- 4.3. Chromosome and genome size evolution, remove this, make as one part as discussion, its my suggestion.

Makes it less structured/clear for the reader in our opinion.

Reviewer 3 Report

The manuscript integrated several approaches to disentangle the origin and evolution of an interesting polyploid complex in Crocus series Verni. The authors found that chromosome numbers in same cases didn’t represent the ploidy level or genome size, partially due to the dynamic variation in karyotypes or multiple origins of dysploidy. They suggested ‘total evidence’ from different aspects, for example, cytotype, phylogeny and morphology, should be employed to untangle the secret of polyploid complex. I found the overall manuscript was well-written and intriguing to broad readers in the field of plant taxonomy and systematics. I only have two main considerations. (1)  Since interspecific hybridization are common within a genus, is there any potential progenitor outside the series Verni also contributing the polyploid speciation? I guess the authors mainly included samples from Verni and didn’t incorporate many more progenitor candidates into their analysis. Therefore, the conclusion of polyploid evolution may not be safely given in case of unsampled relative species in Crocus. (2) As mentioned by Stebbins, ploidy changes from lower to higher levels are intuitively irreversible. I am wondering that it may also hold true for variation in chromosome numbers from less to more in case of even numbers, for examples, from 8 to 10, or from 10 to 12. Is it possible that we could use such information to infer the evolutionary trend of Verni here?

Some other minor comments or suggestion:

(1)    For the part of introduction (e.g., L51), should the authors add a clear definition of polyploidy?  Which academic standard do they use to define polyploidy in this study, duplication of whole genome size or whole set of chromosome number?

(2)    The method used for ITS (e.g., L119), do they obtain the sequences by clone or direct sequencing? It will be helpful to mention here rather than guiding readers to citation.

(3)    The tools used for processing nuclear sequences (e.g., L173) could be updated by newer methods (for example, https://doi.org/10.1111/nph.14111) if possible.

(4)    Previous estimates of genome size should be paid attention due to different method used during each other’s experiments (e.g., L192).

(5)    What a kind of standard used in L198, external or internal?

(6)    L242, how many specific traits were included for leaf section and what were they?

(7)    L275, was it significant or not? How about the correlation coefficient?

(8)    Did the authors date their chloroplast or nuclear trees? It would be informative to see the ages of polyploid lineages in Verni.

Author Response

(1)  Since interspecific hybridization are common within a genus, is there any potential progenitor outside the series Verni also contributing the polyploid speciation? I guess the authors mainly included samples from Verni and didn’t incorporate many more progenitor candidates into their analysis. Therefore, the conclusion of polyploid evolution may not be safely given in case of unsampled relative species in Crocus.

For the Verni group we are quite sure that no taxa from outside the series are involved in polyploid formation. We would see this clearly in the analyzed single-copy genes. Moreover, we are working on a larger study for the entire genus. Results from these analyses indicate monophyly for ser. Verni.

(2) As mentioned by Stebbins, ploidy changes from lower to higher levels are intuitively irreversible. I am wondering that it may also hold true for variation in chromosome numbers from less to more in case of even numbers, for examples, from 8 to 10, or from 10 to 12. Is it possible that we could use such information to infer the evolutionary trend of Verni here?

This is contradicted by the existence of dysploid chromosome number changes, where ascending and descending series exist. Particularly in crocuses.

Some other minor comments or suggestion:

(1)    For the part of introduction (e.g., L51), should the authors add a clear definition of polyploidy?  Which academic standard do they use to define polyploidy in this study, duplication of whole genome size or whole set of chromosome number?

We define it as WGD (first sentence). The main conclusion from our study is that neither genome size nor chromosome number might be enough to detect it.

(2)    The method used for ITS (e.g., L119), do they obtain the sequences by clone or direct sequencing? It will be helpful to mention here rather than guiding readers to citation.

We accidentally included ITS in the Sanger sequencing part in Methods. Indeed sequences were obtained by amplicon sequencing on the Illumina platform, as described for the nuclear single-copy genes. Ribotypes are homogenized except for one C. cf. heuffelianus WCC (Slovakia) individual, where still the C. heuffelianus ITS is present (although only in 15% of the reads).

(3)    The tools used for processing nuclear sequences (e.g., L173) could be updated by newer methods (for example, https://doi.org/10.1111/nph.14111) if possible.

The existing pipelines are using the information of primers and are designed for amplicons, which are not fragmentized i.e. are sequenced as one piece (on PacBio or Nanopore). Our amplicons are digested by an enzyme and many of the fragments don’t have a primer sequence. For orcp with over 1000 bp and a read length of 250 bp these tools won’t work. We can, however, assemble alleles/haplotypes as we sequence fragments of 400 to 600 bp in paired-end mode, so that allele-specific SNPs, which are not further away than the distance covered by such a fragment, can be used for assembly of the marker regions.

(4)    Previous estimates of genome size should be paid attention due to different method used during each other’s experiments (e.g., L192).

We refer for genome size only to our own measurements, conducted on the same machinery, with identical chemistry and standards.

(5)    What a kind of standard used in L198, external or internal?

Always internal.

(6)    L242, how many specific traits were included for leaf section and what were they?

The 20 characters from the relevant literature, related with the leaf blade, leaf mesophyll, epidermal layers and vascular bundles (please see the Supplementary Table S4; the character abbreviation legend is given at the end of the table). The 8 of them were highlighted as the most important ones based on PCA (Supplementary Table S7).  Therefore, these 8 characters were included in CDA (Supplementary Table S8).

(7)    L275, was it significant or not? How about the correlation coefficient?

We added the correlation coefficient in Figure S1. The negative correlation was only significant when C. longiflorus was added in ser. Verni.

(8)    Did the authors date their chloroplast or nuclear trees? It would be informative to see the ages of polyploid lineages in Verni.

We can currently just state that they are very young. However, how young this is we can’t say. It seems that the diploid species in ser. Verni are not older than 1 MY, the polyploids have, accordingly, to be younger. Dating is part of an ongoing analysis for the entire genus.

Round 2

Author Response

Raca et al. aimed at disentangling a polyploid complex within Crocus series Verni, using phylogenetic, phylogenomic, and morphological methods. Crocus is an economically (cultivars), but also an evolutionarily important and interesting genus (hybridization, polyploidy, disploidy). In detail, the authors conducted a multi-approach workflow based on chromosome counts, genome size measurements, genetic/genomic markers (chloroplast markers, nuclear single-copy genes, and GBS), and morphological/anatomical traits data. The authors are experts in these fields, and published many studies within the last ca. 10 years.

They clarified many of the addressed issues, but did not provide some of these explanations in the main text. These explanations are crucial for understanding the methodology of the MS, and thus again a careful revision should take place.

P.1, L. 44-46: People always state that the described workflow of the study is kind of useful. Please be more specific about which kind of analysis/species group it is useful, and how it makes methodological progress.

All. Combined. Single ones don’t help.

→ This is no answer, and please revise your sentence to be more precise

>> Essentially this answer is at the center of our study. We wanted to see what is necessary to resolve evolutionary relationships within this polyploid complex. At the end we arrived at just this: We need all methods because only their combination allows safe conclusions.

We now extended the Abstract to make this clear to everyone.

Moreover, please add GPS coordinates to all tables including location information (e.g., Table 1, S1, S2)

We are generally very reluctant to provide exact origin data, particularly GPS coordinates. Crocuses are nice ornamentals and poaching, resulting in population extinctions, happened several times when we published this information. Therefore we give country and locality but no coordinates. If somebody wants to harvest the plants he/she has to at least travel there and hike around and search the area.

→ Ok, but then add this explanation (as a shortened one) to the main text. It is standard to give GPS coordinates in ecological and evolutionary papers to ensure reproducibility.

>> Done

P.3., L.121: Citation of Geneious is missing. Where are genetic/genomic raw data deposited (deposition is usually mandatory)?

Geneious is distributed through the company Geneious. Thus, it's kind of self-explanatory. Data availability is stated in the “Data Availability Statement”.

→ Yes, but the correct citation of Geneious is: 10.1093/bioinformatics/bts199. Please correct it.

>> Is that correct, as it refers to Geneious Basic that is rather different from today’s program and its developers? We now added Alexei’s 2012 paper instead of the company name.

P.3., L.135-148: You cannot simply take IPYRAD default settings for creating genomic alignments. Settings have a tremendous influence on under- and overmerging of reads into loci (and loci into final alignments; orthologs vs. paralogs) and thus final alignments (number of loci, number of SNPs, missing data percentage, etc., see 10.1002/tax.12365, 10.1111/mec.15919). Please give details about your final ipyrad output filter settings (number of loci, SNPs, missing data), and make sure that different clustering thresholds (you only used 0.85 default) have no tremendous influence on final alignment characteristics and thus phylogenetic results.

We totally agree here! During the course of the study several settings (thresholds, proportion of shared polymorphic sites in a locus etc.) were tested and we also used other tools/approaches (STACKS, an own pipeline similar to ddocent). Output files like the vcf files were checked, reads were also mapped to loci, which had a relatively high number of SNPs, and checked to see if they are potential paralogs (however, diploids were found to be homozygous or if heterozygous then with max. 2 alleles).Finally, the default clustering threshold performed well to recover a sufficient number of loci across the whole series, same for the maximum number of shared heterozygous sites to filter out paralogs. At the end we arrived at the default settings working best (good work by Eaton and co!).

AND

Number of loci, missing data etc. is given, e.g., in 3.3. or partly also in the respectives figures.

In addition, which ploidy setting did you choose? For instance, specifying ploidy = 2 would mean that all alleles above two are filtered out (bad for polyploids). It is also recommended by IPYRAD to do separate parameter optimization for di- and polyploids, and merge both assemblies in IPYRAD step 6. That is due to the fact that different ploidy levels have different genetic similarities within and across loci, and thus affect clustering threshold, assemblies, and final results (see again 10.1002/tax.12365, 10.1111/mec.15919). Please make sure that you handled different ploidy types correctly within analyses.

Most of the SNPs are bi-allelic in the polyploids in our data set (was checked in the vcfs files after running ipyrad with different settings regarding the maximum number of alleles plus by comparing the number of loci as well as by mapping). Also the ddocent similar approach (first generating a pseudo-reference using only diploids, then mapping all samples to that reference) was free assumptions on minimum number of alleles per SNP per loci. Furthermore, only around 2.5% of loci of e.g. C. vernus and C. heuffelianus constituted more than 2 possible alleles over all samples of these species. In our finally used data set, using 2 or 4 for ploidy resulted in the identical number of loci.

AND

  1. 5, L. 173: And which coverage you are aiming at, for diploids and polyploids? The number of reads alone bears not much information.

No, number of height depths cluster i.e. loci produced by the cutting enzymes is needed to calculate the app./average coverage (added). We target a coverage of 30-100x.

→ Ok, but then please add all these information (and the supplemental matrices and figures) to the manuscript. There are no word number restrictions here. Otherwise the readers cannot know that you performed such approaches/analyses to ensure correct read assembly and analyses.

>> Number of high depth clusters is already included in the Suppl. Table S5, their coverage was added. A vcf files was provided as supplementary material. It is mentioned now in the manuscript that we check the allelic constitution.

  1. 5, L. 177-178, and in general: Why did you not use GBS data for reconstructing the evolutionary history of allopolyploids as well? RADPainter+fineRADstructure, or PhyloNetworks can handle GBS data, and are able to inform about reticulate evolution.

Not really well. For example, RADPainter+fineRADstructure didn’t identify the allopolyploids at all (similar to the ancestral population assignment tools used).

These programs claim they can do it, but if you use diverse analysis tools they are not congruent.

→ Yes, RADpainter can be used to infer and discern allopolyploids (e.g., 10.1111/nph.18284). I do not want to force you to take this program, but I aimed at showing different programs able to unravel origin of polyploids. Morevoer, it is not problematic if analysis tools are not showing congruent results, but one have to discuss it because it can be due to biological and/or algorithmic differences.

>> We now include a RADpainter analysis – with the expected result that it can partly detect polyploids (in cases that were already clear from other analyses) and partly not. Did we gain additional insights from the output of the program? Rather not.

  1. 5, L. 200: Please rephrase the entire sentence for better comprehensibility. Did you mean that you did not measure the same individuals within FC and molecular analyses? That’s bad…particularly in a group of varying chromosome numbers and disploidy. In how many cases did the samples overlap? Please provide information on which samples are used in which analyses so that the reader can easily access it.

But we write that we use wherever possible the very same individual for DNA extraction and FC. In rare cases when this was not possible we used several plants from the same population. With this we control for uniformity in the population. Moreover, in all cases we would immediately detect outliers in

the GBS analyses, where individuals from polyploid populations group together separate from the diploid individuals and are characterized by their Ho class. In cases where we had mixed di- and tetraploid populations only FC-measured individuals made it into GBS.

→ Ok, but please give this information in a simple excel sheet so that readers can access it.

>> Information added to Table S3

  1. 6-7: Which morphological traits were evaluated? Are they taxonomically informative? (please cite literature, or add a paragraph to the introduction about which traits are used for taxonomic classification within this group). Moreover, are all traits continuous (if not, how did you treat ordinal/categorical data in PCA/DCA?)? Did you standardize all traits before running PCA? Is a PCA suitable here (e.g., gradient length < 2.5?)? Maybe an evolutionary PCA is more appropriate (https://search.r-project.org/CRAN/refmans/adiv/html/evopca.html ).

We add citations of the relevant literature for this group to M&M part. The total list of characters related with the morphology and leaf anatomy is presented in the Supplementary Table S4 (the abbreviation legend is given at the end of the Table). The qualitative characters were converted to numbers of course. The main reason we used the PCA was just to highlight really significant features we could furthermore use in CDA.

→ Again, please give this information in the main text.

>> The requested information is now included in the main text.

This is a very important issue. According to the intro, we assume allopolyploid relationships, and thus reticulate evolution. Reticulate evolution hurts the assumptions of a phylogenetic tree (i.e., bifurcating patterns). Thus, you need to find another analysis to evaluate all ploidy levels together (e.g., RADpainter, sNMF, etc.). I would thus recommend shifting this tree to the Supplements, or deleting it.

We see in these trees if certain polyploids group together and apart from diploid progenitors or together with them. Particularly MP is rather sensitive to reticulations (hybrids and allopolyploids group normally at the base/as sister to one of their diploid progenitors) and is a good indicator of gene flow. Therefore the methods we use to detect and discern the different groups of tetraploids is valid. Particularly as we don’t claim that these trees reflect the true phylogeny of the group. This we summarize in Fig. 8.

→ You cannot perform tree analysis based on diploid and allopolyploid samples because tree analysis force evolutionary patterns into bifurcations which is simply not true in allopolyploids characterized by network-like evolution. Please give at least 1-2 sentences to clarify careful interpretation of such analyses for allopolyploids in the MS.

>> We added some sentences to explain our rationale for analyzing mixed-ploidy datasets with MP and state once more that the reticulate relationships were provided in Figure 8.

Fig. 2-3. Support values are missing.

These are the schemes illustrating the detailed trees given in the supplements. They provide information on the groups found and how chloroplast types are distributed in the di- and polyploid accessions. Thus, support values in the supplemental materials should be enough.

→ Please state in the legends.

>> Is in

Fig. 6. Explained axis percentages are missing. (b) arrows are missing that indicate in which direction a morphological trait is increasing/decreasing, and factor strength (i.e., length of arrow).

The percentages can be derived from the Supplementary Table S8 (see the last row “Prp. Totl.” - a cumulative proportion - means that first two axes are carrying 85% of the total variability: Root1 55%, Root2 30%. The direction of morphological features can also be seen in Supplementary Table S8 (sign “-” is indicating the negative correlation with specific axis), as well as the factor strength (with a threshold of 0.7).

→ Please add this information to the Figure. Such metrices and arrows also standard. The reader cannot always look up the supplements for interpreting a figure within the main manuscript.

>> Percentages are given in Fig. 6 now.